# Improved Convex Decomposition with Ensembling and Boolean Primitives

## Abstract

Describing a scene in terms of primitives – geometrically simple shapes that offer a parsimonious but accurate abstraction of structure – is an established and difficult fitting problem. Different scenes require different numbers of primitives, and these primitives interact strongly. Existing methods are evaluated by predicting depth, normals and segmentation from the primitives, then evaluating the accuracy of those predictions. The state of the art method involves a learned regression procedure to predict a start point consisting of a fixed number of primitives, followed by a descent method to refine the geometry and remove redundant primitives.

CSG representations are significantly enhanced by a set-differencing operation. Our representation incorporates *negative* primitives, which are differenced from the positive primitives. These notably enrich the geometry that the model can encode, while complicating the fitting problem. This paper demonstrates a method that can (a) incorporate these negative primitives and (b) choose the overall number of positive and negative primitives by ensembling. Extensive experiments on the standard NYUv2 dataset confirm that (a) this approach results in substantial improvements in depth representation and segmentation over SOTA and (b) negative primitives make a notable contribution to accuracy. Our method is robustly applicable across datasets: in a first, we evaluate primitive prediction for LAION images. Code will be released upon acceptance of the paper.

## 1. Introduction

Geometric representations of scenes and objects as *primitives* – simple geometries that expose structure while suppressing detail – should allow simpler, more general reasoning. It is easier to plan moving a cuboid through stylized free space than moving a specific chair through a particular living room. As another example, an effective primitive representation should simplify selecting and manipulating objects in scenes as in image-based scene editing (Bhat et al., 2023; Vavilala et al., 2023). But obtaining primitive representations that abstract usefully and accurately has been hard (review Sec. 2).

Primitive prediction methods for objects are well established (Sec. 2), but are rarer for scenes. There are two main types of method. A **descent method** chooses primitives for a given geometry by minimizing a cost function. Important obstacles include: different geometries require different numbers of primitives; the choice of primitive appears to be important in ways that are opaque; the fitting problem has large numbers of local minima; and finding a good start point is difficult. In particular, incremental fitting procedures are traditionally defeated by interactions between primitives. Sec. 3.3 demonstrates an extremely strong and quite simple descent-based fitting baseline. A **regression method** uses a learned predictor to map geometry to primitives and their parameters. These methods can pool examples to avoid local minima, but may not get the best prediction for a given input. The SOTA method (Vavilala & Forsyth, 2023) for parsing indoor scenes uses a regression method to predict a start point consisting of a fixed set of primitives; this is then polished and redundant primitives removed.

For **negative primitives**, the predicted geometry is the set difference between the union of positive primitives and the union of negative primitives. Admitting negative primitives significantly enriches the range of geometries that can be encoded (Sec. 3.1). This paper shows two procedures that yield significant (over 50% relative error) improvements in accuracy. First, we allow a small number of *negative* primitives in the sense of constructive solid geometry (CSG). Second, we show that selecting the number of primitives per scene (using an appropriately constructed ensembling method) produces very strong improvements in accuracy at

---

[1]Anonymous Institution, Anonymous City, Anonymous Region, Anonymous Country. Correspondence to: Anonymous Author <anon.email@domain.com>.

Preliminary work. Under review by the International Conference on Machine Learning (ICML). Do not distribute.

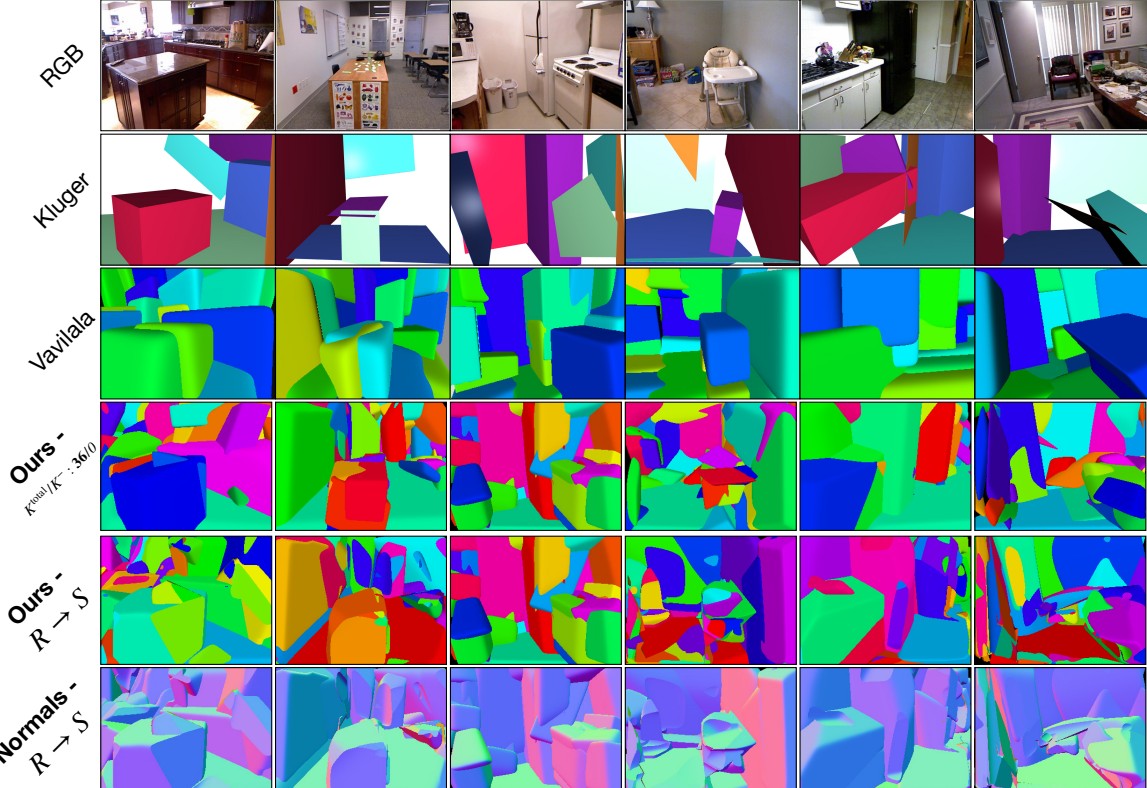

*Figure 1.* We present a method that advances the SOTA for primitive decomposition of indoor scenes by using ensembling and boolean primitives. We present qualitative comparison with prior work here. (**4th row**) We show results from our approach with 36 primitives, none boolean. Our procedure encodes geometry quite closely. (**5th and 6th row**) With boolean primitives, we can encode a rich arrangement of shapes. Here, we ensemble many predictors and show the best one - typically a model with boolean primitives is chosen. The normals make it clear that boolean primitives are scooping geometry away from positive primitives.

small cost in inference time.

Our contributions are:

1. We believe our method is the only one that can fit CSG models including a set differencing operator to in-the-wild images of scenes. We demonstrate qualitative and quantitative benefits to fitting models with negative primitives.

2. Primitive decomposition is unusual, in that one can evaluate a predicted solution at test time without ground truth primitive data by comparing to a depth prediction. We show that this property allows us to search very efficiently for the right number of positive and negative primitives for each scene. The resulting estimate is significantly better than any obtained using a fixed number of positives and negatives.

3. Our method outperforms all available baselines and SOTA on NYUv2, and we demonstrate that it produces accurate representations of diverse scenes from LAION.

## 2. Related Work

Primitives date to the origins of computer vision. Roberts worked with blocks (Roberts, 1963); Binford with generalized cylinders (Binford, 1971); Biederman with geons (Biederman, 1987). Ideally, complex objects might be handled with simple primitives (Chen et al., 2019) where each primitive is a semantic part (Biederman, 1987; Binford, 1971; van den Hengel et al., 2015). Primitives can be recovered from image data (Nevatia & Binford, 1977; Shafer & Kanade, 1983), and allow simplified geometric reasoning (Ponce & Hebert, 1982).

For individual objects, neural methods could predict the right set of primitives by predicting solutions for test data that are "like" those that worked for training data. Tulsiani *et al.* parse 3D shapes into cuboids, trained without ground truth segmentations (Tulsiani et al., 2017). Zou *et al.* parse with a recurrent architecture (Zou et al., 2018). Liu *et al.* produce detailed reconstructions of objects in indoor scenes, but do not attempt parsimonious abstraction (Liu et al., 2022). Worryingly, 3D reconstruction networks might rely

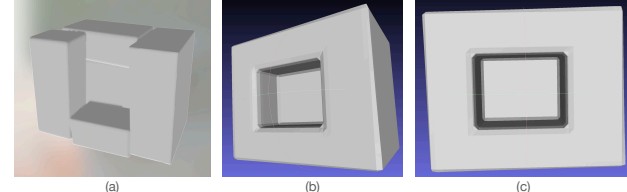

*Figure 2.* **Inference Overview:** An RGBD image is input to an ensemble of independently trained CNNs. Each network predicts the parameters of a set of convexes $\mathcal{C}_i$. The number of convexes varies between 12 and 32 in this work, with many of them potentially being negative. We refine each set of convexes by minimizing the training loss w.r.t. the input depth map. Our final decomposition consists of the set of refined convexes $\mathcal{C}_i$ which yields the lowest absolute relative depth error.

*Figure 3.* **Boolean primitives are parameter-efficient.** Representing a simple box with a hole punched in it can be challenging even with several traditional primitives, as shown in **(a)**, where five primitives get stuck in a local minimum. In contrast, two primitives - one positive and one negative - can represent the geometry successfully because of the enriched vocabulary of operations. Two views are shown in **(b)** and **(c)**.

on object semantics (Tatarchenko et al., 2019). Deng *et al.* (CVXNet) represent objects as a union of convexes, again training without ground truth segmentations (Deng et al., 2020). An early variant of CVXNet can recover 3D representations of poses from single images, with reasonable parses into parts (Deng et al., 2019). Meshes can be decomposed into near convex primitives, by a form of search (Wei et al., 2022). Part decompositions have attractive editability (Hertz et al., 2022). Regression methods face some difficulty producing different numbers of primitives per scene (CVXNet uses a fixed number; (Tulsiani et al., 2017) predicts the probability a primitive is present; one also might use Gumbel softmax (Jang et al., 2017)). Primitives that have been explored include: cuboids (Calderon & Boubekeur, 2017; Gadelha et al., 2020; Mo et al., 2019; Tulsiani et al., 2017; Roberts et al., 2021a; Smirnov et al., 2019; Sun & Zou, 2019; Kluger et al., 2021); superquadrics (Barr, 1981; Jaklič et al., 2000; Paschalidou et al., 2019); planes (Chen et al., 2019; Liu et al., 2018a); and generalized cylinders (Nevatia & Binford, 1977; Zou et al., 2017a; Li et al., 2018). There is a recent review in (Fu et al., 2021).

Neural Parts (Paschalidou et al., 2021) decomposes an object given by an image into a set of non-convex shapes. CAPRI-Net (Yu et al., 2022) decomposes 3D objects given as point clouds or voxel grids into assemblies of quadric surfaces. DeepCAD (Wu et al., 2021) decomposes an object into a sequence of commands describing a CAD model, but requires appropriately annotated data for training. Point2Cyl (Uy et al., 2022) is similar, but predicts the 2D shapes in form of an SDF. Notably, (Yu et al., 2022; Wu et al., 2021; Uy et al., 2022) also utilise CSG with negative primitives or parts but, unlike our work, focus on CAD models of single objects instead of complex real-world scenes.

Hoiem *et al* parse outdoor scenes into vertical and horizontal surfaces (Hoiem et al., 2005; 2007); Gupta *et al* demonstrate a parse into blocks (Gupta et al., 2010). Indoor scenes can be parsed into: a cuboid (Hedau et al., 2009; Vavilala & Forsyth, 2023); beds and some furniture as boxes (Hedau et al., 2010); free space (Hedau et al., 2012); and plane layouts (Stekovic et al., 2020; Liu et al., 2018b). If RGBD is available, one can recover layout in detail (Zou et al., 2017b). Patch-like primitives can be imputed from data (Fouhey et al., 2013). Jiang demonstrates parsing RGBD images into primitives by solving a 0-1 quadratic program (Jiang, 2014). Like that work, we evaluate segmentation by primitives (see (Jiang, 2014), p. 12), but we use original NYUv2 labels instead of the drastically simplified ones in the prior work. Also, our primitives are truly convex. Monnier *et al* and Alaniz *et al* decompose scenes into sets of superquadrics using differentiable rendering, which requires calibrated multi-view images as input (Monnier et al., 2023; Alaniz et al., 2023). Most similar to our work is that of Kluger *et al*, who identify cuboids sequentially with a RANSAC-like greedy algorithm (Fischler & Bolles, 1981; Kluger et al., 2020; 2021; 2024; Kluger & Rosenhahn, 2024).

The success of a descent method depends critically on the start point, typically dealt with using greedy algorithms (rooted in RANSAC (Fischler & Bolles, 1981); note the prevalence of RANSAC in a recent review (Kang et al., 2020)); randomized search (Ramamonjisoa et al., 2022; Hampali et al., 2021); or multiple starts. Remarkably, as Sec. 3.3 shows, modern first-order methods (we used AdamW (Loshchilov & Hutter, 2019)) are capable of producing fairly good primitive representations from a random start point.

## 3. Method

An important feature of this class of problem is that, *at inference time*, one can evaluate a predicted solution efficiently and accurately by comparing primitive predicted depth with a depth map predicted from an image. This makes it possible to polish a representation predicted by a network, and to choose between representations. Write $K^{total}$ for the total

Figure 4. Visualizations of various primitive predictions for four scenes from NYUv2. We show ground truth (first column in each block); predictions of $(12, 0)$, $(24, 0)$ and $(36, 0)$ models; the prediction of the model that made the worst prediction for the scene; and the prediction of the model that made the best prediction. The best choice of primitive numbers varies from scene to scene. Notice some complex objects made up as composites of positive primitives (black arrow) and negative primitives "carving out" shapes. The segmentation label is the oracle label described in the text. Best viewed at high resolution in color.

number of primitives and $K^-$ for the number of negatives to be predicted. For each $(K^{total}, K^-)$ we wish to investigate, we train a prediction network. Then, at inference time, we produce a set of primitives from each network, polish it, select the primitives with the best loss and report that. This means that different scenes will have predictions involving different numbers of primitives. Our approach is a straightforward generalization of the architecture of Vavilala & Forsyth (2023), but produces very significant improvements in performance (Sec. 4). Negative primitives require some minor modifications of their procedure (Sec. 3.1) We use their losses with some omissions (Sec. 3.2). Our polishing procedure is somewhat different (Sec. 3.3).

Our network requires RGBD input. Our losses require a point cloud that is extracted from the depth image via the heuristic described in Vavilala & Forsyth (2023). Our method works both when GT depth is and is not available, as we can use pretrained networks (Ranftl et al., 2022; Yang et al., 2024a) to obtain inferred depth maps. Fig. 2 provides an overview of our inference pipeline.

### 3.1. Positive and Negative Primitives

**Base primitives:** Our primitives are smoothed polytopes as described in (Deng et al., 2020). For 6-faced paral-

lelepipeds, each primitive is parametrized by a center (3 DOF's), 3 normals (6 DOF's), 6 offsets (6 DOFs) and a blending term (1 DOF). The blended half-plane approach eases training and also enables fitting curved surfaces. We fit parallelepipeds/cuboids mainly for fair comparative evaluation, but we show that more faces per polytope yields even better representations (see Table 6 in supplementary).

**Negative primitives:** Set differencing produces a notably more complex geometric representation. Assume we have $K^{total}$ primitives of which $K^-$ are negative, each with $f$ faces. Label an image pixel by the face intersection that produced that pixel (as in our face segmentation figures, e.g. Fig. 5). Generic pixels could result from either ray intersection with a face of a positive primitive or with a face of a negative primitive inside some positive. This argument means that there are a maximum of $f \times (K^{total} - K^-) \times (1 + K^-)$ pixel labels; note how this number grows very quickly with an increase in the number of negative primitives, an effect that can be seen in Fig. 5. Negative primitives are easily handled with indicator functions. We define the indicator for a set of primitives $O : \mathbb{R}^3 \to [0, 1]$, with $O(x) = 0$ indicating free space, and $O(x) = 1$ indicating a query point $x \in \mathbb{R}^3$ is inside the volume. Write $O^+(x)$ for the indicator function for the set of positive primitives, $O^-(x)$

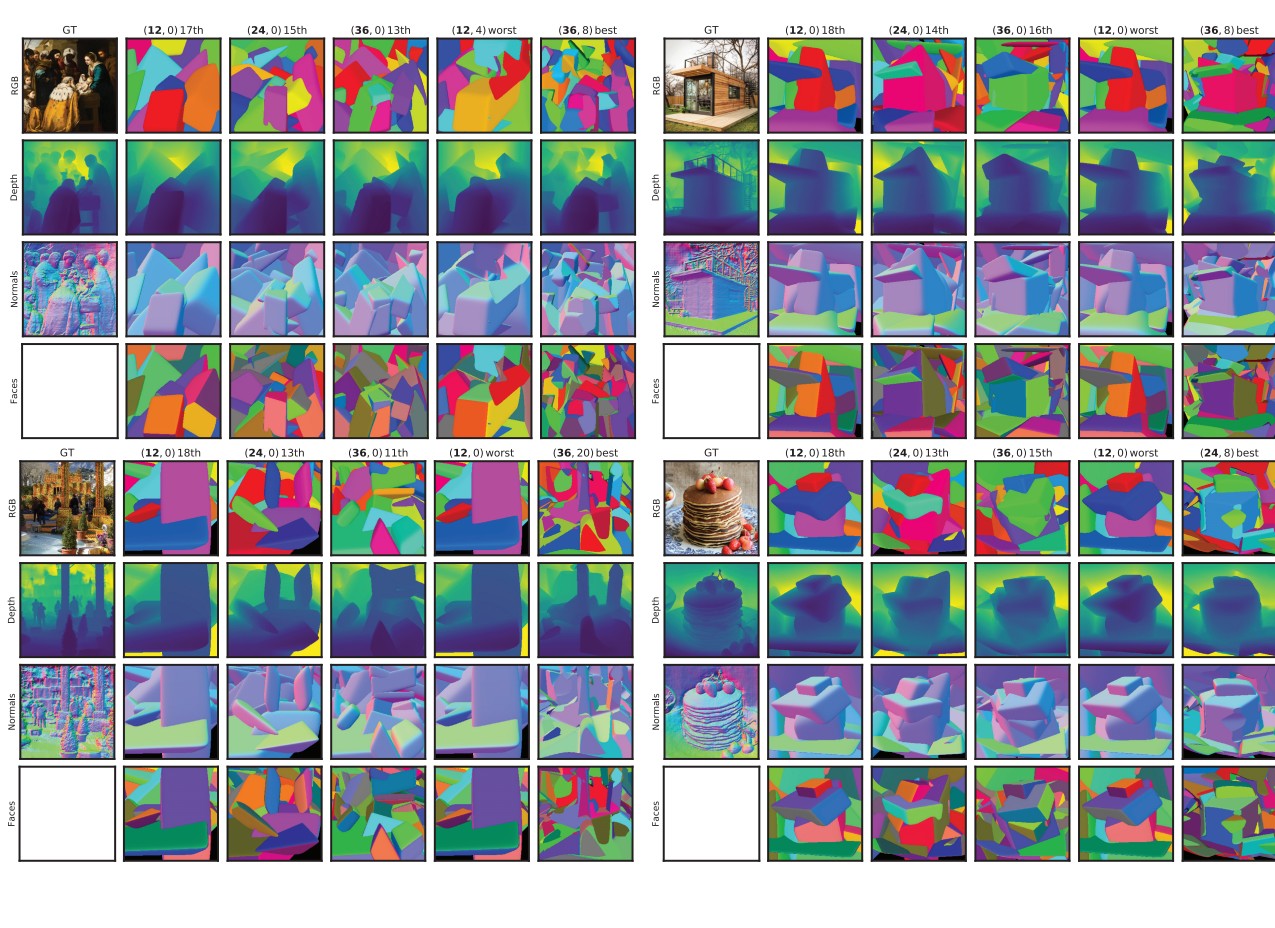

*Figure 5.* Visualizations of various primitive predictions for four scenes from LAION. We show ground truth (first column in each block); predictions of $(12, 0)$, $(24, 0)$ and $(36, 0)$ models; the prediction of the model that made the worst prediction for the scene; and the prediction of the model that made the best prediction. The best choice of primitive numbers varies from scene to scene. **Bottom row** in each block shows face labels – no oracle segmentation is available. Note how primitives can follow complex structures; how they tend to "stick" to object properties (for example, heads in **top left**; the house in **top right**); and how the number of face labels grows very quickly with the number of negative primitives.

for negative primitives. The indicator for our representation is then $O(x) = relu(O^+(x) - O^-(x))$

## 3.2. Losses

Our modified representation allows re-using the existing sample loss and auxiliary losses (unique parametrization loss, overlap loss, guidance loss, localization loss) (Deng et al., 2020; Vavilala & Forsyth, 2023) for both $O^+(x)$ and $O^-(x)$. While a Manhattan World loss was found to be helpful for NYUv2, it hurt quality on general in-the-wild LAION images in our testing so we leave out that loss in this work. We do not consider the volume loss or segmentation loss from Vavilala & Forsyth (2023) in our experimentation, as they were shown to have an approximately neutral effect in the original paper.

## 3.3. Polishing and Descent

Test-time finetuning is possible because we can evaluate the primitive prediction against the predicted depth map, then use the training losses at test-time. The fit is improved by using more 3D samples in these losses per image at test-time. Our polishing procedure has been heavily optimized (Supplementary).

Our polishing procedure is effective enough that we can fit a primitive representation *using only a random start* (details in Supplementary). We are aware of no other primitive fitting procedure that can operate with pure descent and no random restart or incremental process. This supplies an interesting baseline; Sec. 4 demonstrates that this baseline is highly inefficient compared to polishing a network prediction, and is not competitive in accuracy.

| Method | $K^{total}$ | $K^-$ | time (sec) | Memory (GB) | AbsRel ↓ | Normals Mean ↓ | Normals Median ↓ | SegAcc ↑ |
|---|---|---|---|---|---|---|---|---|
| **12**/*0* | 12 | 0 | 0.84 | 3.53 | 0.072 | 36.6 | 30.7 | 0.633 |
| **24**/*0* | 24 | 0 | 1.46 | 5.57 | 0.059 | 35.9 | 29.9 | 0.690 |
| **36**/*0* | 36 | 0 | 2.06 | 7.61 | 0.049 | **34.9** | **29.0** | 0.729 |
| best - **36**/*8* | 36 | 8 | 2.06 | 7.61 | 0.049 | 36.9 | 30.8 | **0.742** |
| pos $S \to R$ | 26.42 | 0.0 | 2.08 | 7.61 | 0.057 | 35.6 | 29.6 | 0.697 |
| pos+neg $S \to R$ | 28.44 | 13.5 | 2.13 | 7.61 | 0.055 | 37.0 | 31.0 | 0.713 |
| pos $R \to S$ | 34.95 | 0.0 | 6.21 | 7.61 | 0.049 | 35.0 | **29.0** | 0.726 |
| pos+neg $R \to S$ | 34.73 | 11.6 | 29.9 | 7.61 | **0.044** | 36.19 | 30.2 | **0.742** |
| (Vavilala & Forsyth, 2023) | 13.9 | 0 | 40.0 | 6.77 | 0.098 | 37.4 | 32.4 | 0.618 |

*Table 1.* Comparison to SOTA (last row) on NYUv2. Our best approach (second last row) polishes then chooses from 18 different models with different numbers of primitives. Other rows show variants of our model. **First three rows**: we train a primitive generation model according to the procedure laid out in Sec 3, without boolean primitives. Next row: 36 total primitives with 8 negative was our best network as measured by AbsRel. **Final four rows** Ensembling strongly improves error metrics, particularly AbsRel. Pos+neg refers to all 18 models available for ensembling, whereas Pos refers to only 3 models without boolean primitives available. $S \to R$ refers to only refining the output of the model with the best sample classification; $R \to S$ means we finetune all models and pick the best one. In this table, we finetune assuming GT depth is available at test time, though our method still works even when depth is inferred by a pretrained depth estimator. The fact that substantial gains can be achieved from $R \to S$ implies that the best start point may not yield the best end point – meaning the fitting problem is hard. Time and memory estimates are presented as well. **Last row**: we compare our methods against existing work. Any individual model we train obtains better error metrics with less compute. Timings for ensembling shows estimated total cost of running all the methods and selecting the best one; memory refers to peak GPU memory usage.

### 3.4. Choosing the Number of Primitives

Much of the literature on primitive decomposition fits a fixed number of primitives (Deng et al., 2020). In contrast, we investigate 18 cases for $(K^{total}, K^-)$. These are $(12, 0)$; $(12, 4)$; $(12, 8)$; $(24, 0)$; $(24, 4)$; $(24, 8)$; $(24, 12)$; $(36, 0)$; $(36, 4)$; $(36, 8)$; $(36, 12)$;$(36, 16)$; $(36, 20)$; $(36, 24)$; $(36, 28)$; and $(36, 32)$. We investigate two strategies for choosing the best prediction (and so the best set of primitives) for a given test image: $S \to R$, where we select the best neural prediction then refine it; and $R \to S$, where we refine all predictions then select the best.

### 3.5. Implementation Details

Our neural architecture is a ResNet-18 encoder (accepting RGBD input), followed by a decoder consisting of three linear layers of sizes $[1048, 1048, 2048]$ and LeakyRelu activations. We do not freeze any layers during training. The dimensionality of the final output varies based on the number of primitives the model is trained to produce (as we train different models for different numbers of primitives in this work). We implement our procedure in PyTorch and train all networks with AdamW optimizer, learning rate $2 \times 10^{-4}$, batch size 96, mixed-precision training, for 20000 iterations, on a single A40 GPU. Each image is resized to $240 \times 320$ resolution. Although we train at fixed resolution, our model can run inference at variable aspect ratio, as would be expected from CNNs like ResNet. It takes 39 mins to train a 12 primitive model and 62 mins to train a 32 primitive model. On LAION, we train at $256 \times 256$ resolution, resizing the smallest edge to 256 and doing a center crop. We increase the training steps to 30000 here, which was sufficient to get good results despite the larger dataset.

## 4. Results

Qualitative results appear in Fig 4 and Fig 5. Note how primitives can combine to form complex structures; how negative primitives "carve out" complex shapes; how primitives tend to "stick" to object properties (for example, heads; a house); and how the number of face labels grows very quickly with the number of negative primitives.

### 4.1. Evaluating a Primitive Representation

While *producing* a primitive representation has a long history (Marr & Nishihara (1978), Sec. 2), not much is known about how one is to be *used* apart from the original recognition argument, now clearly an anachronism. Recent work in conditioned image synthesis (Vavilala et al. (2023); Bhat et al. (2023)) suggests that applications might need (a) a relatively compact representation (so that users can, say, move primitives around) and (b) one that accurately reflects depth, normals and (ideally) segmentation.

We compare primitive methods against one another using standard metrics for depth, normal and segmentation. Specialized predictors of depth, normal and segmentation outperform primitive methods on these metrics. But we would not use a primitive predictor to actually predict depth, normal or segmentation – instead, we are using the metrics to determine whether very highly simplified representations achieve reasonable accuracy. Our procedure uses the standard 795/654 train/test NYUv2 split Nathan Silberman & Fergus (2012). We hold out 5% of training images for vali-

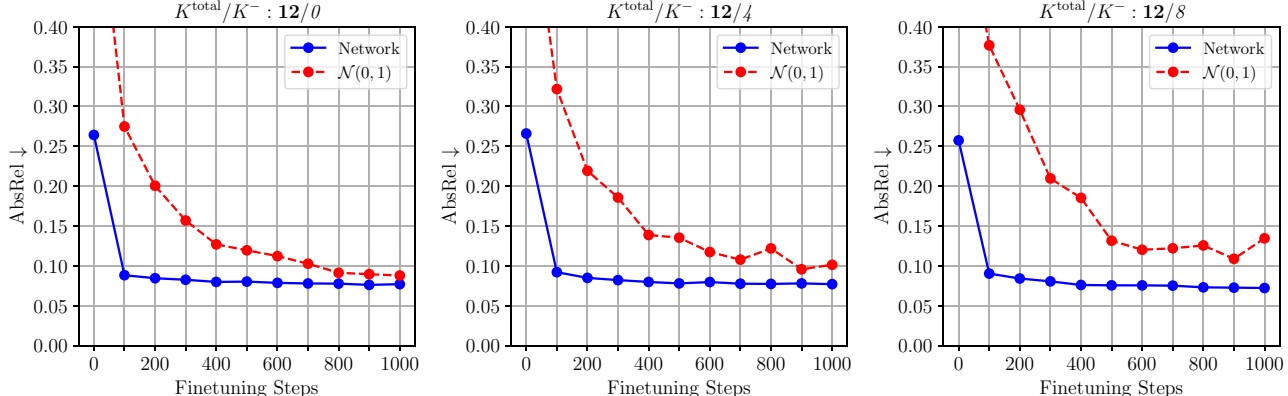

Figure 6. **Network start is beneficial.** Initializing our finetuning process with primitives predicted by our network (blue line) yields better primitives after finetuning than random start (red line). Inference from our network is around 0.0006 seconds per image, whereas 200 iterations of finetuning takes nearly .7 seconds when $K^{total} = 12$. In practice, network start saves around 900 FT steps and can achieve a better quality than random start. It also appears to be harder to fit negative primitives than positives (there is a greater gap in the final AbsRel between the two curves when there are negative primitives). Based on these results, we use 200 finetuning steps to balance compute and quality when reporting error metrics in this paper. Further, the fact that it is even possible to generate high-quality primitives via pure optimization, without a neural network, is new in the context of recent primitive-generation literature. Results shown on 100 random test images from LAION.

dation. We use this dataset primarily to maintain consistency in evaluating against prior art.

For NYUv2, we compare the depth map predicted by primitives to ground truth using a variety of metrics; normals predicted by primitives to ground truth; and an oracle segmentation derived from primitives to ground truth segmentation. Depth metrics are: the (standard) AbsRel (eg (Ranftl et al., 2020)); $AUC_n$, which evaluates the fraction of points within $n$ cm of the correct location (after Vavilala & Forsyth (2023); Kluger et al. (2021)); mean and median of the occlusion-aware distance of (Kluger et al., 2021). Normal metrics are after (Wang et al., 2015) and are mean and median of angle to true normal, in degrees. The oracle segmentation metric uses an oracle to predict the best label for each image region, where regions consist of pixels with the same face intersection label (of Sec. 3.1), then compares this to ground truth. For LAION, we compute depth and normal metrics comparing to depth and normal predicted from the image.

### 4.2. NYUv2 Results

**Our method beats SOTA** on depth, normal, and segmentation (Tab. 1). Despite the ensembling process, **our method is faster than SOTA**. Notice in Tab. 1 methods with no negative primitives show some improvement in normal, but are much worse in depth. We speculate that this is caused by a tendency for real objects to bulge out rather more than to be pressed in. An improvement in depth prediction combined with a weakening of normal prediction is not paradoxical (one can be better at predicting a function and worse at pre-

dicting its derivatives). Qualitative results in Fig. 4. More extensive detailed comparison in Supplementary.

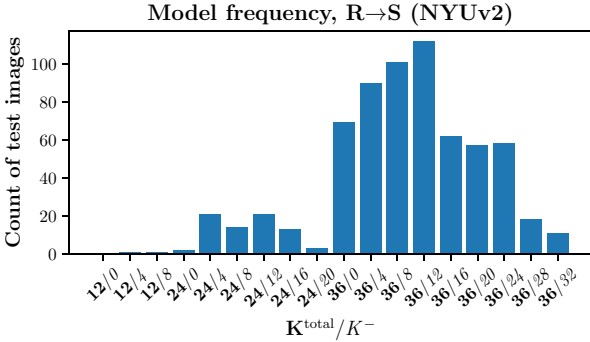

Figure 7. The number of times each primitive model is selected for test images strongly suggests that (a) negative primitives are helpful and (b) bias is not the reason a model is chosen. This figure is for the $R \to S$ strategy, which is best; others in suppplementary.

**Negative primitives make an important contribution**, as indicated by Fig. 7. This figure shows the histogram of the number of times a particular $(K^{total}, K^-)$ combination was selected. Note that there is a strong tendency to use more primitives ($K^{total} = 36$ is much more popular than other options, though $K^- = 24$ is quite popular), and the number of negatives used for the best fit is quite variable.

**Improvements are not just from improved bias**, as Fig. 7 indicates. Generally, a representation with more primitives will have lower bias (one could use one primitive per pixel,

| Dataset | faces | AUC$_{@50}\uparrow$ | AUC$_{@20}\uparrow$ | AUC$_{@10}\uparrow$ | AUC$_{@5}\uparrow$ | mean$_{cm}\downarrow$ | median$_{cm}\downarrow$ |
|---------|-------|-----------|-----------|-----------|----------|---------|-----------|
| NYUv2 | 6 | 0.9391 | 0.8847 | 0.8137 | 0.6902 | 0.128 | 0.03117 |
| LAION | 6 | 0.9097 | 0.838 | 0.7498 | 0.63 | 0.2522 | 0.06239 |
| LAION | 12 | 0.9158 | 0.8502 | 0.7679 | 0.6535 | 0.2366 | 0.05706 |

*Table 2.* Depth metrics for NYUv2 data and LAION data compared. **LAION is harder than NYUv2**, but **more faces yield better fits**.

and get very good depth measurements). But the representations used are quite well spread across different cases in this figure, suggesting that bias is not the issue here (if it were, all representations would be $(36, 24)$).

| $K^{\text{total}}$ | Encode | Loss | Finetune | Render |
|------|--------|------|----------|--------|
| 12 | 0.0006 | 0.0015 | 0.68 | 0.15 |
| 24 | 0.0006 | 0.0025 | 1.23 | 0.22 |
| 36 | 0.0006 | 0.0036 | 1.79 | 0.26 |

*Table 3.* Estimated inference breakdown times, all times in seconds, 256-res images. Encoding is very fast, in which the network predicts parameters of the primitives given an RGBD image. Computing loss, required for getting the fraction of samples classified correctly when ensembling with $S \to R$, is also fast. However, finetuning (we show 200 steps here) is often the bottleneck since we must compute the loss and optimize the parameters of the primitives. Since our primitives are the blended union of halfspaces (Deng et al., 2020), they cannot be rasterized easily and raymarching the SDF is required. We use torch.jit, batching, and pure BFloat16 for all stages of inference except rendering to maximize throughput. We find that rendering must be done in FP32 to avoid unwanted artifacts.

**Our method is efficient**, as Tab. 3 shows. The vast majority of time is spent in polishing the representation.

### 4.3. LAION Results

Scaling is an important concept in computer vision, but we have not seen this concept applied to 3D primitive generation. To that end, we collect approx. 1.8M natural images from LAION-Aesthetic. We use a recent SOTA depth estimation network (Yang et al., 2024b) to obtain depth maps, and make reasonable camera calibration assumptions to lift a 3D point cloud from the depth map. In particular, we use the Hypersim (Roberts et al., 2021b) module that predicts metric depth and use its camera parameters to get the point cloud for each image, which is required for training our convex decomposition model. GT normals can be obtained using the image gradient method described in (Vavilala & Forsyth, 2023), which requires point cloud input. **LAION is harder than NYUv2** as Tab. 2 shows conclusively.

**Our network start is much better than pure descent**, as Fig. 6 shows. The randomly started pure descent procedure of Section 3.3 produces surprisingly strong fits, but requires a large number of iterations to do so. Typically, 100 itera-

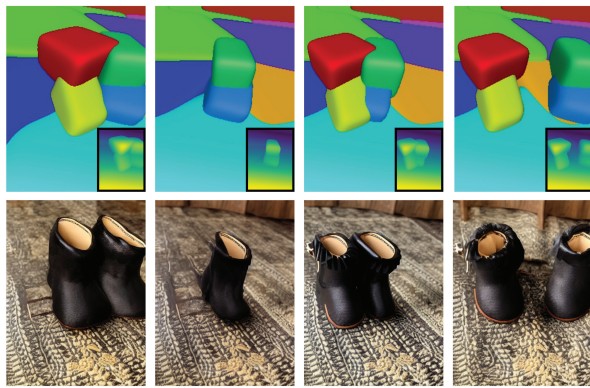

*Figure 8.* **Using primitives to control image synthesis**. We show results from an in-submission follow-up work. Our primitive representation allows us to remove and add objects to a scene. **Bottom row** We generate an image conditioned on primitives (here, primitives extracted from a real image); we then manipulate the primitives and the camera to obtain conditioning for the diffusion model. Depth and primitives shown in **top row**, generated images in second row. Texture is preserved by caching keys and values from a reference style image, and querying those keys and values when generating new images in the same style.

tions of polishing a network start point is much better than 1000 iterations of pure descent. The descent procedure is a first order method, so we expect AbsRel to improve no faster than 1/iterations, suggesting that this figure understates the advantage of the network start point.

## 5. Discussion

Primitives are an old obsession in computer vision. Their original purpose (object recognition) now appears to be much better handled in other ways. Mostly, using primitives was never really an issue, because there weren't viable fitting procedures. But what are primitives for? Likely answers come from robotics – where one might benefit from simplified representations of geometry that are still accurate – and image editing – where a user might edit a scene by moving primitives (Fig. 8).

## Impact Statement

Our work could be useful for high-level scene analysis and understanding. There are many potential societal consequences of our work, none which we feel must be specifically highlighted here.

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

## A. Optimizing the Inference Pipeline

Given the expense of ensembling, we seek to maximize throughput of our inference pipeline. We use torch.jit and pure BFloat16 for encoding the RGBD image and finetuning. We also get speedups from batching the test images instead of one at a time. Combined with our subsampling strategy, these improvements yield over an order of magnitude faster inference than prior work, making ensembling more practical (see Table 1).

We note that rendering the primitives still requires FP32 precision to avoid unwanted artifacts. We accelerate our raymarcher by advancing the step size by 0.8*SDF if it is greater than the step size (we use $0.004$ for large-scale metrics gathering, $0.0001$ for beauty renders). We cannot accelerate by the full SDF because it is an approximation of how far the smoothed primitive boundary is.

## B. Primitives by Descent Alone

We generate a large reservoir of 1M free-space (a.k.a. bbx samples) for each test image. We still generate $H \times W$ "inside" surface samples and "outside" surface samples near the depth boundary respectively, with $\epsilon = 0.02$ units separating these surface samples. We remind the reader that our point clouds are renormalized to approx. the unit cube during training to avoid scale issues. Then during finetuning, we subsample from all available samples at each step, providing a rich gradient analogous to the network training process (though here, we're optimizing the parameters of primitives). We found subsampling $10\%$ of available samples sufficient at each step.

Second, we find that vanilla SGD does not produce usable results; instead AdamW (Loshchilov & Hutter, 2019) was required. We set the initial LR to 0.01, and linearly warm up to it over the first $25\%$ of iterations. We then halve the learning rate once at $50\%$ of the steps and again at $75\%$.

## C. Additional Evaluation

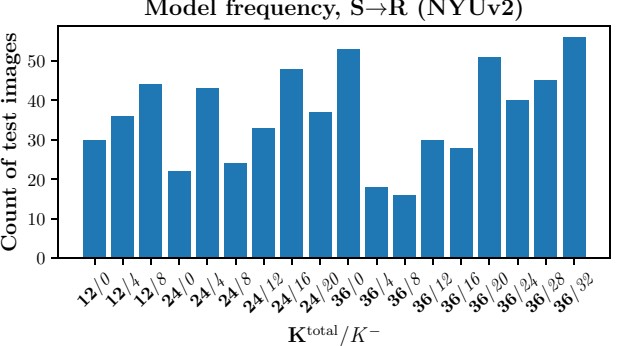

(a) Select then refine ensembling on NYUv2.

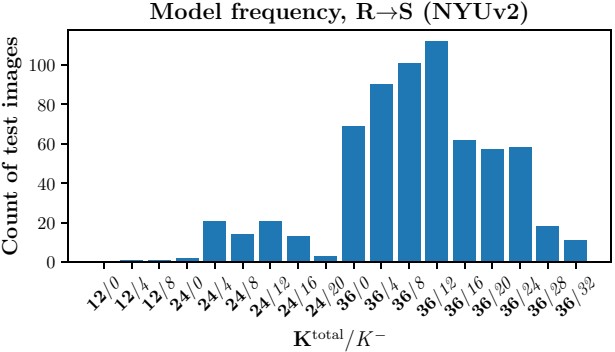

(b) Refine then select ensembling on NYUv2

*Figure 9.* Boolean primitives are often selected from the ensemble. **Top** When we ensemble with $S \rightarrow R$, all models across all primitive counts are well-represented. This indicates that our network prediction may slightly struggle to manage larger numbers of primitives, hence the relative success of fewer primitives. In this setting, selecting a prediction for finetuning is based on fraction of 3D samples classified incorrectly, which is fast as we don't need to finetune and render the outputs of all the networks to decide which model. **Bottom** When we refine then choose $R \rightarrow S$, our finetuning procedure polishes each network start point and chooses the best one based on AbsRel, requiring a render for each model. When doing so, the best model (measured by AbsRel of rendered depth against GT depth) is strongly concentrated among higher primitive counts, $K^{total} = 36$, though fewer primitives are still represented in the ensemble at times. Notice how the ensemble strongly favors representations with boolean primitives available, indicating they are useful in practice.

| Ensemble | Refine | $K^{total}$ | $K^-$ | AUC$_{@50}$↑ | AUC$_{@20}$↑ | AUC$_{@10}$↑ | AUC$_{@5}$↑ | mean$_{cm}$↓ | median$_{cm}$↓ |
|---|---|---|---|---|---|---|---|---|---|
| No (Vavilala 2023) | Yes | 13.9 | 0 | 0.869 | 0.725 | 0.565 | 0.382 | 0.266 | 0.101 |
| No (Kluger 2021) | N/A | - | 0 | 0.772 | 0.627 | 0.491 | 0.343 | 0.208 | - |
| no | yes | 12 | 0 | 0.8967 | 0.8052 | 0.6954 | 0.5411 | 0.2075 | 0.0563 |
| no | yes | 12 | 4 | 0.9102 | 0.8277 | 0.7245 | 0.5737 | 0.1859 | 0.04996 |
| no | yes | 12 | 8 | 0.9044 | 0.8218 | 0.7202 | 0.5712 | 0.194 | 0.04953 |
| no | yes | 24 | 0 | 0.9168 | 0.8485 | 0.7628 | 0.6243 | 0.1697 | 0.04069 |
| no | yes | 24 | 4 | 0.9283 | 0.8685 | 0.7912 | 0.6609 | 0.1479 | 0.03506 |
| no | yes | 24 | 8 | 0.9278 | 0.8667 | 0.7878 | 0.6565 | 0.1497 | 0.03546 |
| no | yes | 24 | 12 | 0.9268 | 0.8652 | 0.7861 | 0.6555 | 0.1519 | 0.03745 |
| no | yes | 24 | 16 | 0.9252 | 0.8601 | 0.7782 | 0.6441 | 0.1552 | 0.0371 |
| no | yes | 24 | 20 | 0.9184 | 0.8445 | 0.7513 | 0.6082 | 0.1713 | 0.04531 |
| no | yes | 36 | 0 | 0.9314 | 0.8751 | 0.8035 | 0.6779 | 0.1408 | 0.03286 |
| no | yes | 36 | 4 | 0.9314 | 0.8755 | 0.8058 | 0.6833 | 0.1395 | 0.032 |
| no | yes | 36 | 8 | 0.9314 | 0.8759 | 0.8073 | 0.6865 | 0.1389 | 0.03134 |
| no | yes | 36 | 12 | 0.9306 | 0.8747 | 0.8061 | 0.6869 | 0.1419 | 0.03391 |
| no | yes | 36 | 16 | 0.9314 | 0.8743 | 0.8037 | 0.6815 | 0.1419 | 0.03184 |
| no | yes | 36 | 20 | 0.9291 | 0.8709 | 0.7974 | 0.6728 | 0.1479 | 0.0373 |
| no | yes | 36 | 24 | 0.9307 | 0.8717 | 0.7947 | 0.6663 | 0.1448 | 0.03595 |
| no | yes | 36 | 28 | 0.9274 | 0.8616 | 0.7791 | 0.644 | 0.1531 | 0.03733 |
| no | yes | 36 | 32 | 0.9244 | 0.8565 | 0.7703 | 0.6321 | 0.1585 | 0.03854 |
| pos | $S \to R$ | 26.42 | 0 | 0.9188 | 0.851 | 0.7672 | 0.6319 | 0.1652 | 0.04028 |
| pos+neg | $S \to R$ | 28.44 | 13.54 | 0.9236 | 0.858 | 0.7765 | 0.6445 | 0.1572 | 0.03789 |
| pos | $R \to S$ | 34.95 | 0 | 0.9316 | 0.8748 | 0.802 | 0.675 | 0.1408 | 0.03344 |
| pos+neg | $R \to S$ | 34.73 | 11.63 | **0.9391** | **0.8847** | **0.8137** | **0.6902** | **0.128** | **0.03117** |

*Table 4.* **Baseline comparisons:** Ensembling strongly outperforms two recent SOTA methods, using the metrics reported by Kluger et al. (2021), and using negative primitives in the ensemble produces further improvements. We show results with only positive primitives present **Ours (pos)**, three networks, $K^{total} \in [12, 24, 36]$, as well as with positive and negative primitives **Ours (pos+neg)**, 18 networks, $K^- \in [0, 4, 8, ..., K^{total} - 4]$. Our ensembles significantly outperform existing work. Further, we present results on the 18 methods we trained, where $K^{total}/K^-$ is shown. Even without ensembling, any individual method we trained performs better than the baselines. Notice that boolean primitives are helpful on average.

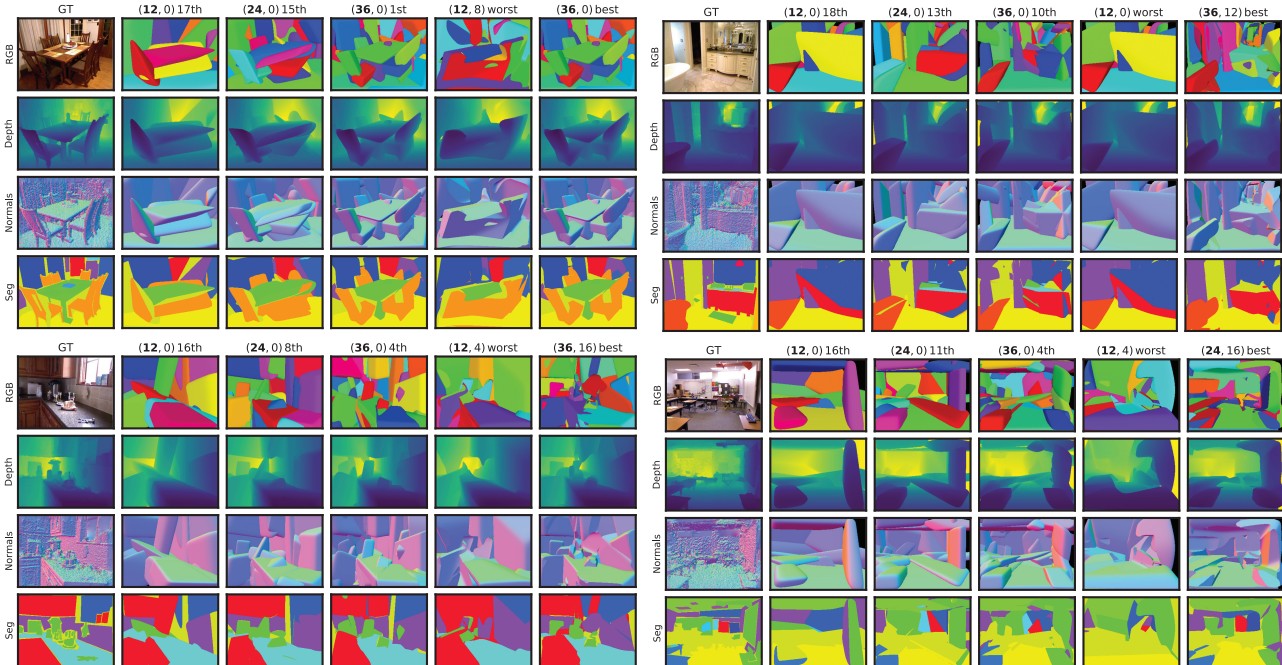

*Figure 10.* Visualizations of various primitive predictions for four scenes from NYUv2, omitting arrows. We show ground truth (first column in each block); predictions of $(12, 0)$, $(24, 0)$ and $(36, 0)$ models; the prediction of the model that made the worst prediction for the scene; and the prediction of the model that made the best prediction. The best choice of primitive numbers varies from scene to scene.

| Ensemble | Refine | $K^{total}$ | $K^-$ | AbsRel↓ | AUC$_{@50}$↑ | AUC$_{@20}$↑ | AUC$_{@10}$↑ | AUC$_{@5}$↑ | mean$_{cm}$↓ | median$_{cm}$↓ | Neg_per_Pos |
|---|---|---|---|---|---|---|---|---|---|---|---|
| no | yes | 12 | 0 | 0.09231 | 0.8698 | 0.7702 | 0.6579 | 0.5197 | 0.3676 | 0.09723 | 0 |
| no | yes | 12 | 4 | 0.09381 | *0.8713* | *0.7727* | *0.6619* | 0.5245 | *0.3581* | *0.09603* | 1.735 |
| no | yes | 12 | 8 | *0.09188* | 0.8707 | 0.7715 | 0.6612 | *0.5248* | 0.3586 | 0.1021 | 4.097 |
| no | yes | 24 | 0 | 0.08226 | 0.8836 | 0.798 | 0.6977 | 0.5693 | 0.3321 | 0.07976 | 0 |
| no | yes | 24 | 4 | 0.07832 | 0.8914 | 0.8116 | 0.717 | 0.5917 | 0.3056 | 0.07286 | 1.26 |
| no | yes | 24 | 8 | *0.07802* | *0.8934* | *0.8152* | *0.7217* | *0.5971* | 0.3024 | *0.07177* | 2.163 |
| no | yes | 24 | 12 | 0.0785 | 0.8927 | 0.8133 | 0.7192 | 0.5943 | *0.2994* | 0.07261 | 3.241 |
| no | yes | 24 | 16 | 0.07812 | 0.8897 | 0.807 | 0.7108 | 0.5852 | 0.3042 | 0.08184 | 4.656 |
| no | yes | 24 | 20 | 0.08493 | 0.8819 | 0.7961 | 0.6967 | 0.5686 | 0.3289 | 0.09769 | 8.119 |
| no | yes | 36 | 0 | 0.0771 | 0.8899 | 0.8103 | 0.7149 | 0.5901 | 0.3173 | 0.07352 | 0 |
| no | yes | 36 | 4 | 0.07642 | 0.8963 | 0.8226 | 0.7336 | 0.6135 | 0.294 | 0.06607 | 1.072 |
| no | yes | 36 | 8 | 0.07392 | 0.8994 | *0.8281* | *0.7413* | *0.6233* | 0.2852 | *0.06513* | 1.688 |
| no | yes | 36 | 12 | 0.07576 | 0.8989 | 0.8274 | 0.7406 | 0.6225 | 0.2854 | 0.06642 | 2.251 |
| no | yes | 36 | 16 | *0.07193* | 0.8997 | 0.8268 | 0.7392 | 0.6209 | 0.2802 | 0.06528 | 2.947 |
| no | yes | 36 | 20 | 0.07346 | *0.8998* | 0.826 | 0.7367 | 0.6168 | *0.279* | 0.06617 | 3.723 |
| no | yes | 36 | 24 | 0.0766 | 0.8952 | 0.8189 | 0.7282 | 0.6069 | 0.2946 | 0.07524 | 4.729 |
| no | yes | 36 | 28 | 0.07655 | 0.8922 | 0.8116 | 0.717 | 0.593 | 0.3002 | 0.08088 | 6.373 |
| no | yes | 36 | 32 | 0.08094 | 0.8872 | 0.8032 | 0.7058 | 0.5797 | 0.3205 | 0.0942 | 9.969 |
| yes | yes | 24.21 | 0 | 0.08427 | 0.8807 | 0.7927 | 0.6902 | 0.5594 | 0.3406 | 0.08586 | 0 |
| yes | yes | 27.87 | 13.8 | 0.07993 | 0.8889 | 0.8069 | 0.7098 | 0.5837 | 0.3097 | 0.07555 | 3.862 |
| yes | yes | 30.76 | 0 | 0.07285 | 0.8936 | 0.8119 | 0.7144 | 0.5884 | 0.3029 | 0.07412 | 0 |
| yes | yes | 34.28 | 13.26 | **0.06025** | **0.9097** | **0.838** | **0.7498** | **0.63** | **0.2522** | **0.06239** | 2.882 |

*Table 5.* **Quantitative evaluation on LAION 6 face polytopes:** We train and ensemble models on a subset of LAION, with approx. 1.8M images in the training set and 2500 in the test set. We report error metrics defined in by Kluger et al. (2021). Negative primitives remain useful, noting the italiciced error metrics in each block of $K^{total}$ always has boolean primitives. Ensembling produces further improvements similar to NYUv2. Overall, the metrics are worse on LAION, indicating it is a harder dataset. The final column, Neg_per_pos, evaluates the average number of negative primitives touching each positive primitive, quantitatively showing negative primitives active in the geometric abstraction.

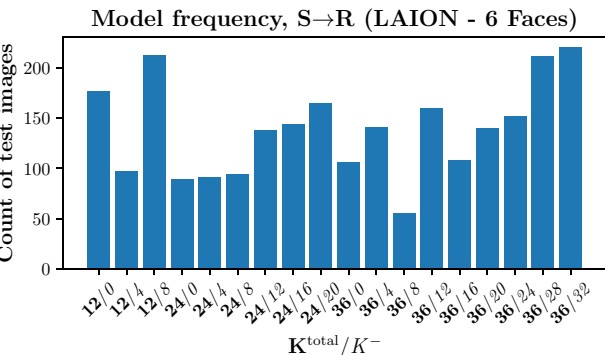

(a) Select then refine ensembling on LAION 6 faces.

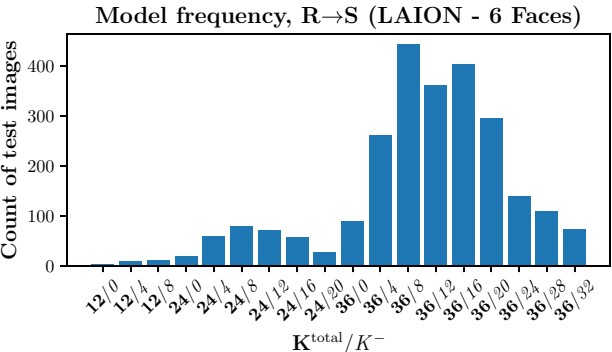

(b) Refine then select ensembling on LAION 6 faces

*Figure 11.* Distribution of models chosen on LAION (6 faces), 2500 image test set. Models with boolean primitives are often chosen, especially after finetuning.

| Ensemble | Refine | $K^{total}$ | $K^-$ | AbsRel$\downarrow$ | AUC$_{@50}\uparrow$ | AUC$_{@20}\uparrow$ | AUC$_{@10}\uparrow$ | AUC$_{@5}\uparrow$ | mean$_{cm}\downarrow$ | median$_{cm}\downarrow$ | Neg_per_Pos |
|---|---|---|---|---|---|---|---|---|---|---|---|
| no | yes | 12 | 0 | 0.08453 | 0.8804 | 0.7854 | 0.6775 | 0.5419 | 0.3331 | 0.09617 | 0 |
| no | yes | 12 | 4 | 0.08138 | 0.89 | 0.8046 | 0.7046 | 0.5753 | *0.3058* | *0.07838* | 1.875 |
| no | yes | 12 | 8 | *0.07951* | *0.8905* | *0.8059* | *0.7063* | *0.5771* | 0.3062 | 0.07857 | 4.149 |
| no | yes | 24 | 0 | 0.07068 | 0.8983 | 0.8206 | 0.7258 | 0.6024 | 0.287 | 0.07173 | 0 |
| no | yes | 24 | 4 | 0.07199 | 0.9025 | 0.8286 | 0.7392 | 0.6185 | 0.2728 | 0.06691 | 1.34 |
| no | yes | 24 | 8 | *0.06978* | *0.9042* | *0.8328* | *0.745* | *0.6259* | *0.2679* | *0.06231* | 2.208 |
| no | yes | 24 | 12 | 0.07175 | 0.9031 | 0.8311 | 0.7438 | 0.6255 | 0.2696 | 0.06407 | 3.333 |
| no | yes | 24 | 16 | 0.07279 | 0.9005 | 0.8264 | 0.737 | 0.6166 | 0.2765 | 0.06635 | 4.725 |
| no | yes | 24 | 20 | 0.07331 | 0.8994 | 0.8244 | 0.7337 | 0.6132 | 0.2822 | 0.06863 | 8.627 |
| no | yes | 36 | 0 | 0.06937 | 0.9023 | 0.8297 | 0.7398 | 0.6208 | 0.2768 | 0.06693 | 0 |
| no | yes | 36 | 4 | 0.06873 | 0.9066 | 0.8384 | 0.7542 | 0.6386 | 0.2628 | 0.05955 | 1.108 |
| no | yes | 36 | 8 | 0.06587 | *0.9091* | 0.842 | 0.7593 | 0.6451 | *0.2536* | 0.05777 | 1.724 |
| no | yes | 36 | 12 | *0.06555* | 0.9091 | *0.8433* | *0.7624* | *0.6494* | 0.2559 | *0.05711* | 2.332 |
| no | yes | 36 | 16 | 0.06708 | 0.9073 | 0.8398 | 0.7574 | 0.6445 | 0.2603 | 0.05879 | 3.023 |
| no | yes | 36 | 20 | 0.0667 | 0.9066 | 0.8387 | 0.7557 | 0.642 | 0.261 | 0.05907 | 3.807 |
| no | yes | 36 | 24 | 0.06928 | 0.9047 | 0.8356 | 0.7509 | 0.6357 | 0.2673 | 0.0621 | 4.952 |
| no | yes | 36 | 28 | 0.06776 | 0.9049 | 0.8339 | 0.7474 | 0.6308 | 0.2655 | 0.06252 | 6.713 |
| no | yes | 36 | 32 | 0.07276 | 0.9007 | 0.8264 | 0.7368 | 0.6174 | 0.2752 | 0.06432 | 10.93 |
| yes | yes | 23.64 | 0 | 0.07552 | 0.8934 | 0.8107 | 0.7127 | 0.5866 | 0.3001 | 0.07877 | 0 |
| yes | yes | 27.2 | 13.23 | 0.07115 | 0.9011 | 0.827 | 0.7373 | 0.6171 | 0.2768 | 0.06681 | 4.096 |
| yes | yes | 31.18 | 0 | 0.06514 | 0.9047 | 0.8308 | 0.7395 | 0.6196 | 0.2685 | 0.06761 | 0 |
| yes | yes | 34.08 | 12.94 | **0.05647** | **0.9158** | **0.8502** | **0.7679** | **0.6535** | **0.2366** | **0.05706** | 2.981 |

*Table 6.* **Quantitative evaluation on LAION 12 face polytopes:** Most recent literature on primitive-fitting focuses on cuboids or parallelepipeds, but our model is capable of fitting polytopes of variable face count. All error metrics get better with more faces, which is helpful to know.

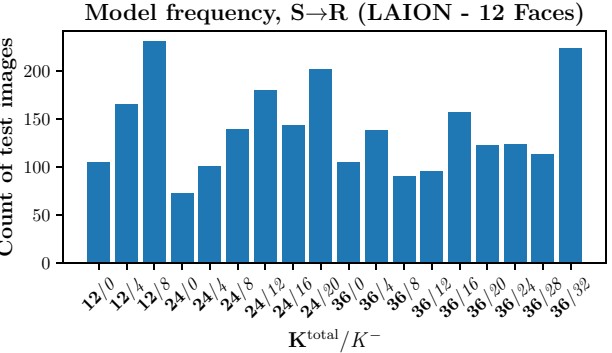

(a) Select then refine ensembling on LAION 12 faces.

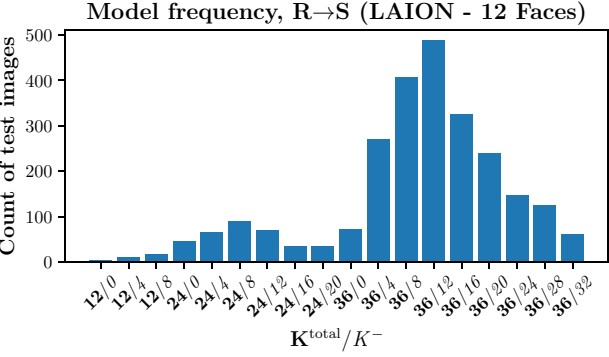

(b) Refine then select ensembling on LAION 12 faces

*Figure 12.* Distribution of models chosen on LAION (12 faces), 2500 image test set. Models with boolean primitives are often chosen, especially after finetuning.

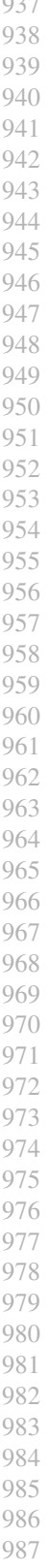

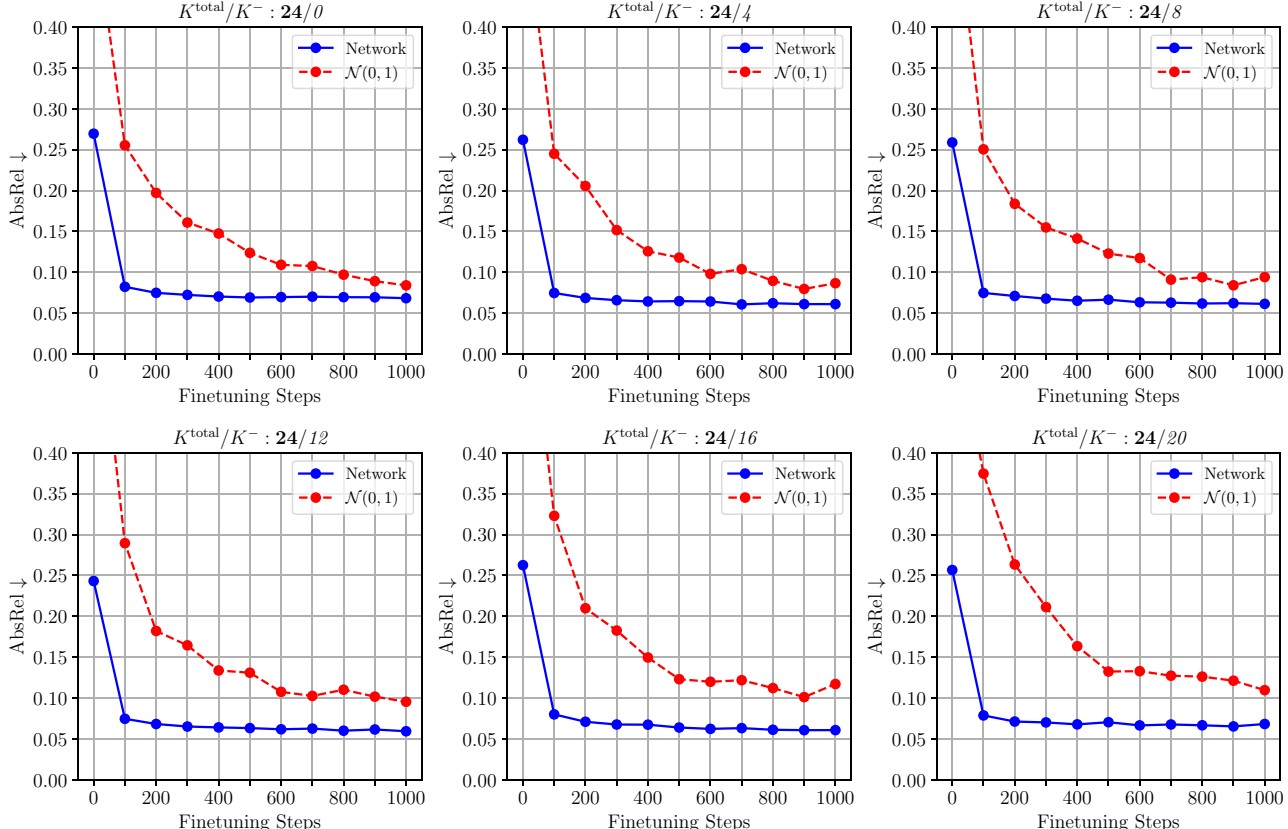

*Figure 13.* Additional examples on the value of network start, on 100 LAION test images.

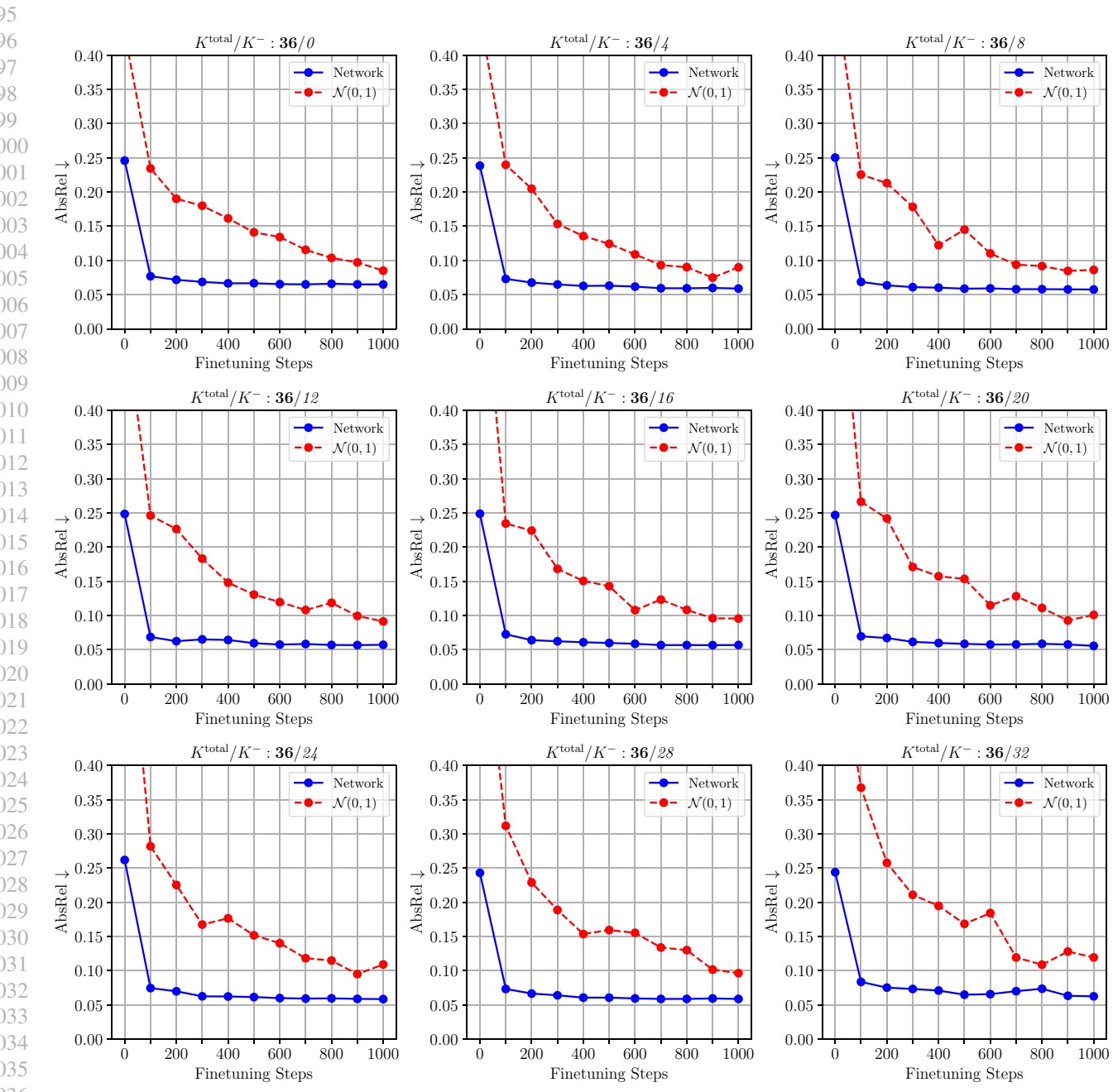

*Figure 14.* Additional examples on the value of network start, on 100 LAION test images.

| Method | NParts | Neg | AbsRel↓ | Normals Mean↓ | Normals Median↓ | SegAcc↑ |
|---|---|---|---|---|---|---|
| Single 12-0 | 12 | 0 | 0.0719 | 36.6 | 30.7 | 0.633 |
| Single 12-4 | 12 | 4 | 0.0659 | 38.2 | 32.1 | 0.656 |
| Single 12-8 | 12 | 8 | 0.0669 | 38.8 | 32.8 | 0.656 |
| Single 24-0 | 24 | 0 | 0.0590 | 35.9 | 29.9 | 0.690 |
| Single 24-4 | 24 | 4 | 0.0525 | 36.3 | 30.4 | 0.719 |
| Single 24-8 | 24 | 8 | 0.0525 | 37.4 | 31.3 | 0.722 |
| Single 24-12 | 24 | 12 | 0.0529 | 37.1 | 31.3 | 0.720 |
| Single 24-16 | 24 | 16 | 0.0538 | 37.7 | 31.6 | 0.714 |
| Single 24-20 | 24 | 20 | 0.0586 | 38.4 | 32.3 | 0.693 |
| Single 36-0 | 36 | 0 | 0.0489 | **34.9** | **29.0** | 0.729 |
| Single 36-4 | 36 | 4 | 0.0496 | 36.4 | 30.4 | 0.737 |
| Single 36-8 | 36 | 8 | 0.0489 | 36.9 | 30.8 | 0.742 |
| Single 36-12 | 36 | 12 | 0.0500 | 36.9 | 31.0 | **0.743** |
| Single 36-16 | 36 | 16 | 0.0497 | 36.6 | 30.5 | 0.740 |
| Single 36-20 | 36 | 20 | 0.0509 | 37.0 | 31.0 | 0.733 |
| Single 36-24 | 36 | 24 | 0.0508 | 37.2 | 31.3 | 0.735 |
| Single 36-28 | 36 | 28 | 0.0528 | 37.3 | 31.2 | 0.720 |
| Single 36-32 | 36 | 32 | 0.0544 | 37.7 | 31.7 | 0.707 |
| pos $S \to R$ | 26.4 | 0.0 | 0.0571 | 35.6 | 29.6 | 0.697 |
| pos+neg $S \to R$ | 28.4 | 13.5 | 0.0546 | 37.0 | 31.0 | 0.713 |
| pos $R \to S$ | 35.0 | 0.0 | 0.0486 | 35.0 | 29.0 | 0.726 |
| pos+neg $R \to S$ | 34.7 | 11.6 | **0.0438** | 36.2 | 30.3 | 0.742 |

*Table 7.* **Detailed error metrics on NYUv2.**

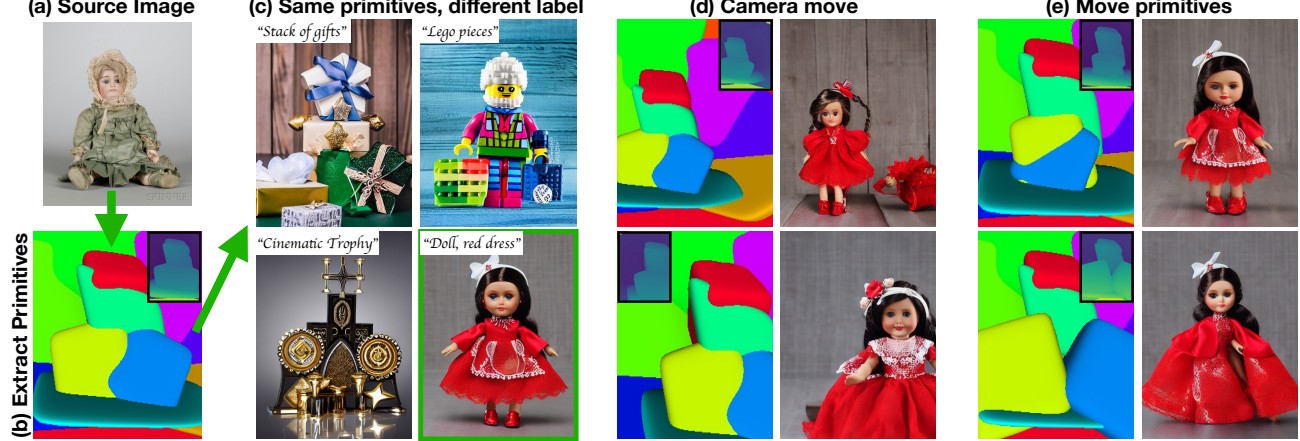

*Figure 15.* **Our method can decompose natural images into primitives, and be used to condition controlled image synthesis tasks**. We show results from an in-submission follow-up work, which uses a convex decomposition method similar to the one described here. **(a)** We use a convex decomposition method to extract convex polytopes from any image. **(b)** We then ray-march the primitives from the original camera viewpoint to obtain a depth map. **(c)** This depth map serves as conditioning to a ControlNet diffusion model, which is finetuned to handle the unique statistics of our block arrangements. Different scenes can be created from the same high-level geometry. **(d)** We can select one of the images and perform camera moves in 3D space, obtaining images that roughly respect both the requested geometric layout and source texture. We maintain a key-value cache to transfer texture (Khachatryan et al., 2023). **(e)** We can also move primitives freely in 3D space, adjusting the high-level shape of the doll's dress.

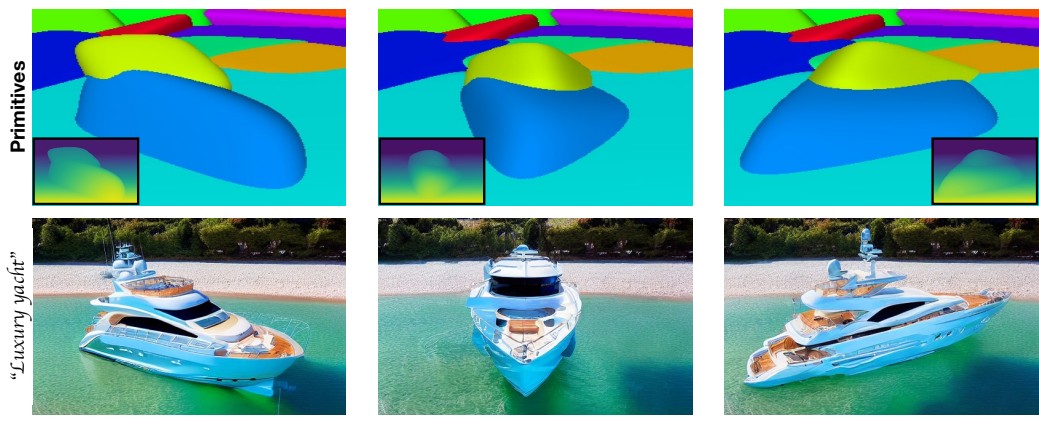

*Figure 16.* **Our method can decompose natural images into primitives, and be used to condition controlled image synthesis tasks**. We show results from an in-submission follow-up work. Rotating the primitives associated with the yacht rotates the yacht in view.

