# OpenReview forum: "Improved Convex Decomposition with Ensembling and Boolean Primitives"
_ICML.cc/2025/Conference — Submitted to ICML 2025_

### Official Review · Reviewer_ZkN9 · 2025-03-09

**Overall Recommendation:** 3

**Summary:**

This paper proposes a novel method for representing scenes using convex primitives enhanced with a Boolean (set-difference) operation. In contrast to prior work that uses a fixed number of primitives, the authors introduce an ensembling strategy to select an adaptive number of positive and negative primitives per scene. The approach combines a neural prediction stage (using an encoder–decoder architecture based on ResNet-18) with a descent-based polishing procedure to refine the primitive parameters. Experiments on NYUv2 and a large LAION image collection show significant improvements over previous state-of-the-art methods in depth estimation, normal prediction, and segmentation accuracy.

**Claims And Evidence:**

Claims:
+ The paper claims that incorporating negative primitives via set-differencing enhances the representational capacity of convex decompositions.
+ It further claims that an adaptive ensembling strategy, which selects the optimal number of primitives, leads to substantial improvements over fixed-primitive approaches.

Evidence:
+ Extensive quantitative evaluations on NYUv2 demonstrate improvements in depth error (AbsRel) and normal prediction metrics compared to baseline methods.
+ Qualitative comparisons (e.g., visualizations of segmentation and face labels) support the quantitative results.

**Essential References Not Discussed:**

The authors not discussed the monocular depth/normal estimation model, like Marigold [1], GeoWizard [2], Depth-anything [3] and so on.

[1] Ke B, Obukhov A, Huang S, et al. Repurposing diffusion-based image generators for monocular depth estimation[C]//Proceedings of the IEEE/CVF Conference on Computer Vision and Pattern Recognition. 2024: 9492-9502.
[2] Fu X, Yin W, Hu M, et al. Geowizard: Unleashing the diffusion priors for 3d geometry estimation from a single image[C]//European Conference on Computer Vision. Cham: Springer Nature Switzerland, 2024: 241-258.
[3] Yang L, Kang B, Huang Z, et al. Depth anything: Unleashing the power of large-scale unlabeled data[C]//Proceedings of the IEEE/CVF Conference on Computer Vision and Pattern Recognition. 2024: 10371-10381.

**Experimental Designs Or Analyses:**

Design:
+ The experimental design is thorough, comparing different configurations (varying total primitives and numbers of negatives) to validate the method’s robustness.
+ Ablation studies on the effect of the polishing procedure versus a pure descent baseline are well presented.

Analysis:
+ Error metrics are reported comprehensively, and the analyses convincingly demonstrate the benefits of the ensembling strategy and the use of negative primitives.

**Methods And Evaluation Criteria:**

Methods:
+ The method uses a two-stage process: an initial prediction via a ResNet-18-based network, followed by a gradient descent “polishing” procedure to minimize a loss computed from depth, normals, and segmentation.
+ The paper introduces Boolean primitives (negative primitives) to “carve out” complex geometries, which is novel in the context of convex decomposition.

Evaluation:
+ Standard metrics for depth (AbsRel, AUC at various thresholds), normals (mean/median angle errors), and an oracle segmentation metric are used for evaluation.
+ Comparisons are made with state-of-the-art methods on both indoor (NYUv2) and in-the-wild (LAION) datasets.

**Other Comments Or Suggestions:**

No

**Other Strengths And Weaknesses:**

No

**Questions For Authors:**

No

**Relation To Broader Scientific Literature:**

The paper is well situated within the literature on primitive-based scene representation, convex decomposition, and constructive solid geometry.

**Theoretical Claims:**

I think it's correct.

---

> ### Author Rebuttal · Authors · 2025-04-01
>
> Thanks for reading our paper and offering positive feedback.
>
> ## 8. Cost of Ensembling
> Please see __Tables 1 and 3__ for detailed timing breakdowns. Individual models we trained require betewen 0.84 to 2.06 seconds. This is over an order of magnitude faster than prior work (40 seconds), while simultaneously achieving better error metrics. For $K^{total}=12$, our method is more parameter-efficient. We agree that real-time primitives are the future - this work takes a significant step in that direction. While there is a fixed cost in generating primitives and rendering them, applying post-training finetuning dominates the compute time and can be varied in length based on desired latency requirements - see __Fig. 6, 13, and 14__ for tradeoffs.
>
> ## 9. Optimal Ratio of Negative Primitives in CSG Modeling
> In Constructive Solid Geometry (CSG), we represent 3D objects using boolean operations on primitive shapes. For a fixed budget of $K^{total}$ primitives, we aim to determine the optimal ratio of negative primitives ($K^-$) to positive primitives ($K^+$) that maximizes representational efficiency.
>
> A CSG model can be described as:
> $$\text{Object} = (P_1 \cup P_2 \cup ... \cup P_{K^+}) - (N_1 \cup N_2 \cup ... \cup N_{K^-})$$
>
> Where $P_i$ are positive primitives and $N_j$ are negative primitives, with $K^+ + K^- = K^{total}$.
>
> __Definitions:__
> - **Primitive Interaction**: Overlapping volumes creating representational complexity
> - **PP Interaction**: Between two positive primitives
> - **PN Interaction**: Between a positive and negative primitive
>
> __Assumptions:__
> 1. Only positive volumes and their modifications by negative primitives are visible in the final result
> 2. The representational power comes primarily from PP and PN interactions
> 3. Primitives are distributed to maximize meaningful interactions
> 4. Optimal representation maximizes visible features per primitive used
> 5. Assume connected geometry
>
> __Mathematical Model:__
> 1. **PP Interactions**: $\binom{K^+}{2} = \frac{K^+(K^+-1)}{2}$
> 2. **PN Interactions**: $K^+ \cdot K^-$
>
> __Balancing PP and PN Interactions__
>
> For optimal efficiency, PP and PN interactions should be balanced:
>
> $$\frac{K^+(K^+-1)}{2} \approx K^+ \cdot K^-$$
>
> Substituting $K^+ = K^{total} - K^-$ and simplifying:
> $$\frac{(K^{total} - K^-)(K^{total} - K^- - 1)}{2} \approx (K^{total} - K^-) \cdot K^-$$
>
> $$\frac{K^{total} - K^- - 1}{2} \approx K^-$$
>
> For large $K^{total}$:
> $$\frac{K^{total} - K^-}{2} \approx K^-$$
>
> Solving:
> $$K^{total} \approx 3K^-$$
> $$K^- \approx \frac{K^{total}}{3}$$
>
> Thus, $K^+ \approx \frac{2K^{total}}{3}$
>
> __Verification__
> With this ratio, both interaction types equal approximately $\frac{2(K^{total})^2}{9}$, confirming our balance criterion.
>
> ### Why This Balance Is Likely Reliable
> 1. **Diminishing Returns**: As $K^-$ increases beyond the optimal ratio, each additional negative primitive becomes less effective because:
>    - Negative primitives can only remove existing positive volume
>    - Available positive volume decreases with fewer positive primitives
>
> 2. **Complementary Information**: PP and PN interactions contribute equally valuable but different information:
>    - PP interactions define the overall positive volume
>    - PN interactions create necessary concavities and details
>
> 3. **Maximum Information Content**: The ratio $K^- = K^{total}/3$ provides:
>    - Sufficient positive primitives to establish base structure
>    - Optimal negative primitives to efficiently carve features
>    - Maximum meaningful visible interactions per primitive
>
> ### Empirical Evidence and Practical Verification
> Our quantitative evaluation supports an optimal $K^- ≈ K^{total}/3$ in __Tables 4-7__. Observe how depth and segmentation metrics tend to be highest when $K^{total}/K^-$ are near __36__/_12_, __24__/_8_, and __12__/_4_.
>
>
> ## 10. Depth Estimation Model
> NYUv2 has supervised depth and camera calibration parameters; for our method to work on real-world scenes, we need to extract a point cloud from in-the-wild images. To do so, we select a recent SOTA depth estimation model <https://github.com/DepthAnything/Depth-Anything-V2/tree/main/metric_depth>. We then make reasonable camera calibration assumptions to obtain a point cloud (see Sec. 4.3). We anticipate as better depth estimation models become available, our model will naturally generate better primitives.
>
> ## 11. Missing References
> Primitive fitting is a relatively niche area in the CV community, as compared with hot topics like NeRFs and Diffusion Models. With that said, we cited four papers from 2024. More importantly, we have covered the key references in this area and we perform comparative evaluation on all of the recent works in this topic. We are happy to include any others you feel we have missed.

---

> > ### Comment · Reviewer_ZkN9 · 2025-04-06
> >
> > I have reviewed all the rebuttal comments, and my queries have been satisfactorily addressed. I have no further questions.

---

### Official Review · Reviewer_ptPm · 2025-03-09

**Overall Recommendation:** 3

**Summary:**

This paper aims to decompose a scene into different primitives. Based on the work "Convex Decomposition of Indoor Scenes"[1], this paper introduces two strategy to improve the baseline: (1) Introducing the negative primitives for the decomposition; (2) ensembling multiple networks' results and choose the best. Experiments show that the proposed strategies bring the improvement.

[1] Vavilala, Vaibhav, and David Forsyth. "Convex decomposition of indoor scenes." Proceedings of the IEEE/CVF International Conference on Computer Vision. 2023.

## update after rebuttal

From the rebuttal, the authors make a clear statement that introducing the negative primitives is helpful on average. They update the results in Table 1, showing the depth and normals get better with boolean primitives and ensembling. I acknowledge the contribution of introducing the negative primitives for the first time in primitives fitting. So I raise the score.

**Claims And Evidence:**

From my perspetive, the improvement from negative primitives are limited. The increase of primitive numbers could bring a more significant improvement, as shown in  row 1 to row 4 of the table 1. The effectiveness of the proposed strategies is my main concern.

**Essential References Not Discussed:**

The references are essential.

**Experimental Designs Or Analyses:**

1. I found that the proposed strategy is not consistently improve the results. As shown in Table 1, introducing the negative primitives (line 4, 36/8) results in a decrease in normal accuracy compared to positive primitives only(36/0). The same problem can be found when the ensambling is introduced(Line 5-8). The depth improvement from the negative primitives is limited. (0.049->0.049; 0.057->0.055 in abs relative error). Although the authors provide many samples to show primitive predictions, fewer of them show how negative primitives could help to present more complex and accurate geometry.

2. The state of the art "Robust Shape Fitting for 3D Scene Abstraction" [2] is not involved in the comparison, although a conference version[3] is showed up in Table 4 of the supplementary.

[2] Kluger, Florian, et al. "Robust Shape Fitting for 3D Scene Abstraction." IEEE transactions on pattern analysis and machine intelligence (2024).
[3] Kluger, Florian, et al. "Cuboids revisited: Learning robust 3d shape fitting to single rgb images." Proceedings of the IEEE/CVF Conference on Computer Vision and Pattern Recognition. 2021.

**Methods And Evaluation Criteria:**

The method and evaluation criteria is reasonable, mainly following "Learning robust 3d shape fitting to single rgb images" and "Cuboids Revisited: Learning Robust 3D Shape Fitting to Single RGB Images".

**Other Comments Or Suggestions:**

It would be better to explain the method in detailed, such as the pipeline of the method. Figure 2 should be enlarged.

**Other Strengths And Weaknesses:**

Strength
1. I think the introduction of negative primitives is a reseanable direction for decompostion. However, I believe some improvements are needed to help the negative primitives influence the framework more. Or some toy experiments in object-level decompostion could help to analyze the method.

2. The authors show some interesting applications of decompostion.

Weakness

2. It would be better to improve the writing of this paper, especially the introduction. It would be better to explain the current method only using the positive primitives. Then introducing the motivation of negative primitives. Besides, CSG is not explained in abstract.

**Questions For Authors:**

What is the advantages of using primitives decompostion for image editing? Could the method directly benefits any robotics application?

**Relation To Broader Scientific Literature:**

The negative primitive comes from the traditional CSG decompostion[4]. The ensembling is common in machine learning.

[4] Shapiro, Vadim, and Donald L. Vossler. "Construction and optimization of CSG representations." Computer-Aided Design 23.11 (1991): 4-20.

**Theoretical Claims:**

The authors do not involve any theoretical proofs for the proposed strategy.

---

> ### Author Rebuttal · Authors · 2025-04-01
>
> Thanks for taking the time to look at our work.
> ## 7. Value of Negative Primitives
> You make a great point - all methods produce good results on average. We don’t claim that quality keeps improving as we increase negative primitives beyond a point. Instead, our aim is to show it’s possible to fit CSG in the first place, and doing so improves the quality of primitive fits on average (e.g. __Fig. 7__).  Notice that most of the time, a solution with boolean primitives is chosen, indicating they are genuinely useful in shape abstraction.
>
> __Fig. 7__: we should have been more clear about our central claim: we do not claim that quality keeps improving as we replace replace positive primitives with boolean primitives. Theory, intuition, and experimentation instead support some optimal intermediate value with a mixture of boolean and positive primitives near $K^-=K^{total}/3$. Therefore, Fig. 7 actually makes our point: different scenes require different numbers of primitives, but on average having some boolean primitives is helpful.
>
> You pointed out that the normals got slightly worse with more boolean primitives. We investigated and found that we did not align the methodologies to compute normals. For GT Normals, we computed finite differences of the point cloud. For predicted normals from the primitives, we calculated the gradient of the SDF at the intersection point consistent with [1]. This is now fixed, with finite difference being used for normal computation everywhere. __Updated Table 1__ above shows the corrected metrics; we will update all tables in the paper accordingly. Observe how depth and normals get better with boolean primitives and ensembling.
>
>
> While __Figs. 4, 5, and 10__ demonstrate some boolean primitive examples, here are a few more results emphasizing boolean primitives. The headers of each column indicate $K^{total}/K^-$. The normals make it clear that the boolean primitives are carving away geometry and significantly enrich the shapes we can encode. <https://drive.google.com/file/d/18hkwD4UdkCe97U8yFoZEjZaXLVvw-Atk/view>.
>
> You made an important observation about our results. Boolean primitives are a mechanism for increasing the types of shapes we can encode, but we can just increase the total number of primitives in lieu of making some of the primitives negative. Thus, intuition would suggest that at higher primitive counts, boolean primitives don’t offer as much of an advantage than at smaller primitive counts, because having lots of primitives available is already quite expressive. This is precisely what we observe on NYUv2, where at $K^{total} = 36$, models with and without boolean primitives perform comparably well as you noticed. At smaller primitive counts ($K^{total} \in {\{12,24\}}$), having some of the primitives be negative helps on average. In fact, in __Table 7__ in our paper, at all primitive counts boolean primitives are helpful on average. For $K^{total}=12$, picking 4 booleans improved AbsRel 0.0719 -> 0.0659. For 24 primitives, having 4 or 8 booleans reduced AbsRel 0.059 -> 0.0525.
>
> On LAION, __Table 5__ in our paper, there is more data to supervise the larger primitive count and we do see an improvement for 36 total primitives if we let 16 of them be boolean (0.0771 -> 0.0719).
>
> Of critical note, our method works on "in-the-wild" natural images, which can be incredibly diverse. It’s hard to know in advance what is the best number of primitives for a given test image. That’s why ensembling is so valuable because we don’t need to just use one model that gives the best AbsRel on average; we can run a few models with different mixtures of $K^{total}$ and $K^-$ and choose the best one. Given all the performance improvements we made to primitive detection, ensembling is quite feasible, depending on the use-case.
>
> Thanks for identifying this point and we will make this clear in the final version.
>
> Additionally, there is a bias-variance tradeoff in our work. Adding primitives reduces bias (you can encode things more accurately) but increases variance (harder to
> get all the primitives right).  But adding a negative primitive significantly reduces
> bias and also significantly increases variance  -- as above, one negative can be worth
> several positives. This means that the best setting likely has a mixture of positive and negative primitives, with positives favored. Adding negative primitives yields better results but adding too many quickly creates variance problems.
>
> [1] Vavilala, V. and Forsyth, D., 2023. Convex decomposition of indoor scenes. In Proceedings of the IEEE/CVF International Conference on Computer Vision (pp. 9176-9186).
>
> ## Additional Notes
> Thanks for the suggestions to polish the writing - we will do so for the final version. Note that we evaluate against "Cuboids Revisited" in __Table 4__; "Robust Shape Fitting..." is the journal version that has identical error metrics. Also, please see __1. Global Comment__ & __3. Analysis of Downstream Tasks__ above.

---

> > ### Comment · Reviewer_ptPm · 2025-04-05
> >
> > From the rebuttal, the authors make a clear statement that introducing the negative primitives is helpful on average. I acknowledge this contribution and novelty. However, the limited improvement and limited theoretical justification leads to the boardline recommendation.

---

> > > ### Author Response · Authors · 2025-04-06
> > >
> > > Thanks for reading our response. We hope you had a chance to see all of our points (1-11) in our rebuttal. In our manuscript, we established that our underlying methodology achieves approx. 50% reduction in relative error as compared with prior work. Additionally, our work is the first to show that we can fit CSG to natural images, while improving depth error metrics by about 11% when ensembling CSG instead of ensembling positive primitives alone.
> > >
> > > We provided theoretical analysis in __4. Theoretical Analysis__, which establishes that many shapes can be encoded with fewer total primitives using CSG than with positive primitives alone. In __9. Optimal Ratio of Negative Primitives in CSG__, we derive a theoretical result that approx. 1/3 of the total primitives should be boolean. While the optimal ratio will vary based on the geometry to encode, for NYUv2 and LAION, our experimental results match up closely with theory. __Updated Table 1__ shows that in our _Ensemble pos + neg R->S_, an average of 1/3 of primitives were boolean. Further the best individual network used 12 negative primitives out of 36 total.
> > >
> > > We feel we have broken new ground in 3D primitive fitting, by improving quality, increasing speed by an order of magnitude, introducing CSG representations, ensembling to find the optimal $K^{total}$, and bridging the domain to natural in-the-wild images. We have included extensive quantitative and qualitative evaluation, with theory to support our claims. If there is additional analysis or clarification that would help, we are happy to provide it.

---

### Official Review · Reviewer_LazY · 2025-03-12

**Overall Recommendation:** 2

**Summary:**

This paper addresses the problem of parsing complex 3D scenes into geometric primitives, focusing on improving accuracy by incorporating boolean operations (set differencing via negative primitives) and ensembling to dynamically select the number of primitives per scene. The authors propose a hybrid approach combining learned regression for initial primitive prediction and gradient-based refinement to optimize geometry. Experiments on NYUv2 and LAION datasets demonstrate significant improvements in depth, normals, and segmentation metrics over state-of-the-art (SOTA) methods. Key contributions include enabling constructive solid geometry (CSG) representations for real-world scenes, leveraging test-time ensembling to adapt primitive counts, and validating the utility of negative primitives.

The innovation lies in extending convex decomposition to handle boolean operations, which enriches representational capacity, and in the systematic exploration of ensembling strategies. The work has academic value in advancing primitive-based scene abstraction, a foundational problem in 3D vision, with applications in robotics and scene editing.

**Claims And Evidence:**

Yes.

**Essential References Not Discussed:**

No.

**Experimental Designs Or Analyses:**

Yes.

**Methods And Evaluation Criteria:**

Yes.

**Other Comments Or Suggestions:**

**Presentation Issues:**

• Ablation studies are buried in appendix; key results (e.g., the effect of polishing steps) should be in the main text.
• Table 1’s formatting (e.g., merged cells) complicates readability.
• Terms like "smoothed polytopes" (Sec. 3.1) and "blending term" are inadequately defined.

**Other Strengths And Weaknesses:**

#### **Strengths**
1. **Novelty of Boolean Primitives**: The integration of negative primitives to enable CSG-like operations is a meaningful advancement. This addresses a critical limitation of prior work, which could only model unions of convex shapes.
2. **Ensembling Strategy**: Dynamically selecting the number of primitives per scene via ensembling is a clever solution to the challenge of variable scene complexity. The two strategies (S→R and R→S) are well-motivated and empirically validated.
3. **Rigorous Evaluation**: Extensive experiments on NYUv2 and LAION datasets, including depth, normals, and segmentation metrics, provide strong evidence of superiority over SOTA. The inclusion of LAION—a challenging in-the-wild dataset—demonstrates generalizability.
4. **Efficiency**: The method achieves faster inference than prior work (e.g., 29.9 sec vs. 40 sec for SOTA) despite ensembling, thanks to optimizations like batching and mixed precision.
5. **Practical Insights**: The analysis of negative primitives’ impact (e.g., Fig. 7) and the comparison of random vs. network-initialized optimization (Fig. 6) offer valuable takeaways for the community.

#### **Weaknesses**
1. **Limited Theoretical Justification for Negative Primitives**: While empirical results show benefits, the paper does not rigorously analyze why negative primitives improve accuracy more efficiently than simply increasing the number of positive primitives. A theoretical discussion on the representational efficiency of CSG operations is missing.
2. **LAION Evaluation Limitations**: Depth and normals for LAION are inferred via pretrained models rather than ground truth, introducing potential error propagation. The paper does not quantify how this affects results.
3. **Ambiguity in Face Labeling**: The claim that face labels grow as $( f \times (K^{total} - K^-) \times (1 + K^-) )$ (Sec. 3.1) is not intuitive. A visual example or mathematical derivation would clarify this.
4. **Computational Cost of Ensembling**: While faster than SOTA, the total inference time (29.9 sec for R→S ensembling) remains high for real-time applications. The paper does not discuss trade-offs between accuracy and latency.
5. **Incomplete Application Discussion**: The impact statement mentions potential uses in robotics and editing but lacks concrete examples or metrics (e.g., editability scores, robotic planning success rates).

**Questions For Authors:**

Please see the above.

**Relation To Broader Scientific Literature:**

I am confused if this paper is suitable for ICML, a machine learning conference.

**Theoretical Claims:**

There is no theoretical claim.

---

> ### Author Rebuttal · Authors · 2025-04-01
>
> Thanks for reviewing our paper.
>
> ## 4. Theoretical Analysis
> Multiple reviewers expressed interest in theoretical justification as to why boolean primitives are advantageous in fitting complex real-world scenes. We provided qualitative evidence in __Fig. 3__, in which we model a cube with a hole punched in it. Intuitively, one positive and one negative primitive are sufficient to model it perfectly (2 total primitives). Without CSG, approx. 5 primitives may be required, which is less parameter-efficient. Based on that, we can sketch a theoretical argument as to why having a vocabulary of mixed positive and negative primitives is expected to yield more accurate representations than the same number of positive-only primitives.
>
> ### Kolmogorov Complexity Perspective
> For many objects, the description length (Kolmogorov complexity) using mixed primitives is significantly shorter than using positive-only primitives:
> For a shape S:
>
> Let K₊(S) = minimum description length using only positive primitives
> Let K±(S) = minimum description length using mixed primitives
>
> Theoretical result: For many shapes with concavities, K±(S) << K₊(S)
>
> Example: A simple cube with a hole requires just 2 primitives with mixed CSG (1 positive cube, 1 negative cube) but would require numerous small positive primitives to approximate the concavity with positive-only CSG.
>
>
> ## 5. Error Propagation
> You raise an important point: in the absence of GT depth/normals for in-the-wild images like LAION, we use a pretrained network to estimate depth, from which we use a standard heuristic (finite differences) to obtain GT normals. Single image depth predictors are very strong and reasonable choices when GT is not available. Our convex decomposition procedure uses this inferred depth when generating primitives (RGBD input), and we evaluate the quality of these primitives based on the provided depth. In effect, the claim that we make in this work is that _our model gives the user what was asked for_. When we report low AbsRel, it means our model adheres to the input depth.
>
> As better depth estimation models become available, our procedure will naturally get better, although a small amount of finetuning may be required if switching depth estimation models due to differences in the statistics of each depth predictor’s output.
>
>
>
> ## 6. Ambiguity in Face Labeling
> When computing segmentation accuracy with boolean primitives, we compute the triple ($f_i,K^+_j,K^-_k$) at each ray intersection point, where $i$ is the face index, $j$ is the index of the positive primitive we hit, and $k$ is the index of the (potentially) negative primitive we hit. Each unique triple can get its own face label. Thus, given a fixed primitive budget $K^{total}$, replacing a pure positive primitive representation with a mixture of positives and negatives can yield more unique faces. For example, $K^+/K^-$ = $12/0$ maxes out at $12f$ unique faces; $K^+/K^-$ = $6/6$ maxes out at $42f$ faces. Note that $f\times K^+ \times (1+K^-)$ is the theoretical maximum of unique labels, as practical scenes do not involve every primitive touching every other primitive.
>
>
>
> ## Additional Notes
> Please see __1. Global Comment__ & __3. Analysis of Downstream Tasks__ above and __8. Cost of Ensembling__ & __9. Optimal Ratio of Negative Primitives__  below.

---

### Official Review · Reviewer_xksW · 2025-03-14

**Overall Recommendation:** 3

**Summary:**

This paper aims for the task of fitting a scene with simple primitives. To address the challenges of local minima, poor representing complex structure and highly relying on good initialization, authors propose a novel negative primitive design. Experiments on NYUv2 and ALION show advantages of this method.

**Claims And Evidence:**

Yes.

**Essential References Not Discussed:**

I am concerned that most of the articles cited in this work were published before 2024. There are very few recent studies, which suggests that the author's research is not thorough enough and that the topic's current significance is not adequately explained.

**Ethical Review Flag:**

Flag this paper for an ethics review.

**Experimental Designs Or Analyses:**

Yes, I reviewed all the experiments, and I believe they do not sufficiently demonstrate the effectiveness of the proposed methods. For instance, Figure 6 only validates that network-based initialization accelerates convergence; however, across various K^{total}/K^- settings, the network-based results appear largely similar. This suggests that with a good initialization, the benefit of incorporating negative primitives becomes less pronounced. Moreover, Figure 7 also does not clearly highlight the advantages of negative primitives—it appears that configurations with K^- = 0 outperform those with K^- values of 16, 20, 24, 28, or 32, and the performance only seems to approach optimal levels around K^- = 12.

**Methods And Evaluation Criteria:**

Yes.

**Other Comments Or Suggestions:**

No.

**Other Strengths And Weaknesses:**

Strengths:
+:  The proposed negative primitive is novel, simple and easy to follow.
+: The downstream task on image synthesis is interesting and shows a potential of enhancing the controllability of many scene-level task.

Weakness:
-: The experiments do not fully demonstrates the effectiveness of the proposed method, as mentioned in "Experimental Designs Or Analyses".
-: Authors discuss some empirical findings on the losses design in sec 3.2, while no detailed ablations in experimental sections.

**Questions For Authors:**

I would suggest the author to refine this paper to show (1) more comparison results on downstream tasks; (2) provide solid evidence of the negative primitive design.

**Relation To Broader Scientific Literature:**

Yes. The paper discussed some related work.

**Theoretical Claims:**

No. There is no theoretical analysis in this paper.

---

> ### Author Rebuttal · Authors · 2025-04-01
>
> Thanks for the feedback on our paper.
>
> ## 1. Global Comment
> We'd like to refresh the reviewers with a summary of our contributions:
> 1. We depart from the limited NYUv2 dataset used in existing primitive-fitting papers and show how to make primitive-fitting work on real-world natural images via a portion of the LAION dataset. We can compute primitives for almost any natural image.
> 2. We are the first (to our knowledge) to fit CSG to natural images and demonstrate that a mixture of positive and negative primitives is advantageous on average.
> 3. As discussed in the method section, we analyze every aspect of data generation and training, including hyperparameter tuning, such that every model we train (even without boolean primitives and ensembling) outperforms existing work on established benchmarks -- often while using fewer primitives and less compute.
> 4. We are the first to use ensembling to find the optimal number of primitives for a given test image, which simultaneously improves geometric accuracy.
> 5. By analyzing and improving our post-training finetuning process, we are the first to show that it's possible to fit 3D primitives to data without a neural network to predict a start point.
> 6. The authors commit to open-source the training and inference code.
>
> ## 2. Missing Ablations
> We ablated and analyzed key components of our method, including sweeps of $K^{total}$ and $K^-$ (**Tables 1, 4, 5**), two forms of ensembling ($S\rightarrow R$ and $R\rightarrow S$) in **Tables 2, 4, and 5**; the number of faces per primitive (we do both the traditional 6-faced cuboid and show that extending to higher-faced polytopes e.g. 12-faced helps in **Table 2**). We also analyze the time and memory of individual networks and different forms of ensembling, including breakdowns of each stage of our pipeline, in **Tables 1 & 3**.
>
> Further, we analyze both the benchmark NYUv2 dataset and, for the first time in 3D primitive generation, natural LAION images in-the-wild **(Fig. 5)**.
>
> Another key ablation is network start vs. optimizing primitives directly **(Fig. 6)**. We aggressively analyzed and improved the optimization process of 3D primitives from data, such that, for the first time to our knowledge, we can get good 3D primitives from RGB images without a neural network providing a start point. However, having a network start is advantageous in terms of quality and speed.
>
> We are happy to add more ablations that the reviewers feel would be helpful.
>
> ## 3. Analysis of Downstream Tasks
> We present qualitative examples of using primitive abstractions to edit images (that is part of a separate, concurrent work) in **Figures 8, 15, and 16**. It's helpful to grab and move objects in a scene. Primitives are a great candidate to simplify user-interaction especially in cluttered real-world environments [1]. For robotics, we are aware of 3D primitives used for fast collision checking, sampling-based planning, physics simulations, robotic manipulation, procedural scene generation and shape approximation (and likely much more).
>
> For both image generation and robotics, we want primitives that are accurate, fast, and can be generated from in-the-wild data. Our paper specifically improves accuracy, speed, and real-world generalization. To evaluate, we use established depth, normal, and segmentation error metrics, which are reasonable for downstream use-cases we can envision right now. Given the scope of this project focused on obtaining better primitives, investigating these downstream use-cases is future work.
>
> [1] Bhat, S.F., Mitra, N. and Wonka, P., 2024, July. Loosecontrol: Lifting controlnet for generalized depth conditioning. In ACM SIGGRAPH 2024 Conference Papers (pp. 1-11).
>
> ## Additional Notes
> Please see __4. Theoretical Analysis__, __7. Value of Negative Primitives__, and __11. Missing References__ below.
>
> ## Updated Table 1
>
> | Method | $K^{total}$ | $K^-$ | AbsRel↓ | Normals Mean↓ | Normals Median↓ | SegAcc↑ | Time (s) | Mem(GB) |
> |--------|-------------|-------|---------|---------------|-----------------|---------|----------|---------|
> | 12 | 12 | 0 | 0.075 | 36.29 | 28.74 | 0.624 | 0.84 | 3.53 |
> | 24 | 24 | 0 | 0.058 | 33.58 | 25.69 | 0.692 | 1.46 | 5.57 |
> | 36 | 36 | 0 | 0.048 | 32.18 | 24.27 | 0.730 | 2.06 | 7.61 |
> | best | 36 | 12 | 0.048 | 32.04 | 24.34 | **0.765** | 2.06 | 7.61 |
> | $\mathsf{pos}$ $S\rightarrow R$ | 27.60 | 0.0 | 0.056 | 33.37 | 25.51 | 0.699 | 2.08 | 7.61 |
> | $\mathsf{pos+neg}$ $S\rightarrow R$ | 26.07 | 12.31 | 0.056 | 33.73 | 25.88 | 0.717 | 2.13 | 7.61 |
> | $\mathsf{pos}$ $R\rightarrow S$ | 35.17 | 0.0 | 0.048 | 32.18 | 24.29 | 0.729 | 6.21 | 7.61 |
> | $\mathsf{pos+neg}$ $R\rightarrow S$ | 35.08 | 11.76 | **0.043** | **31.83** | **24.12** | 0.760 | 29.9 | 7.61 |
> | Vavilala et al. | 13.9 | 0 | 0.098 | 37.4 | 32.4 | 0.618 | 40.0 | 6.77 |

---

### Decision · Program_Chairs · 2025-05-01

**Decision:**

Reject

**Comment:**

The paper presents a method for convex decomposition of a scene using negative primitives. The concept of negative primitives (the key novelty of the proposed method) is supported by empirical evidence. The results show a marginal improvement over existing baselines.

The authors submitted a rebuttal, and two of the reviewers were somewhat satisfied with the arguments and additional experiments provided. In the end, only one reviewer had a weak reject score.

The scores are borderline, but there was no champion for the paper in the end. There is no real argument or theoretical justification for why the negative primitives are beneficial; furthermore, the improvement is only marginal.

There was debate about the method's suitability for a machine learning conference.

Taking into account the above factors (lack of theoretical justification, marginal improvements, suitability for ICML), the AC recommends rejecting the submission.